# Compressing the Activation Maps in Deep Neural Networks and Its Regularizing Effect

**Minh H. Vu**  *minh.vu@umu.se*
*Department of Radiation Sciences*
*Umeå University*

**Anders Garpebring**  *anders.garpebring@umu.se*
*Department of Radiation Sciences*
*Umeå University*

**Tufve Nyholm**  *tufve.nyholm@umu.se*
*Department of Radiation Sciences*
*Umeå University*

**Tommy Löfstedt**  *tommy.lofstedt@umu.se*
*Department of Computing Science*
*Umeå University*

**Reviewed on OpenReview:** *https://openreview.net/forum?id=s1qh12FReM*

## Abstract

Deep learning has dramatically improved performance in various image analysis applications in the last few years. However, recent deep learning architectures can be very large, with up to hundreds of layers and millions or even billions of model parameters that are impossible to fit into commodity graphics processing units. We propose a novel approach for compressing high-dimensional activation maps, the most memory-consuming part when training modern deep learning architectures. The proposed method can be used to compress the feature maps of a single layer, multiple layers, or the entire network according to specific needs. To this end, we also evaluated three different methods to compress the activation maps: Wavelet Transform, Discrete Cosine Transform, and Simple Thresholding. We performed experiments in two classification tasks for natural images and two semantic segmentation tasks for medical images. Using the proposed method, we could reduce the memory usage for activation maps by up to 95%. Additionally, we show that the proposed method induces a regularization effect that acts on the layer weight gradients. Code is available at https://github.com/vuhoangminh/Compressing-the-Activation-Maps-in-DNNs.

## 1 Introduction

In recent years, deep learning (DL) has emerged as a very powerful machine learning (ML) technique. The rapid development of deep neural networks (DNNs) is partly due to the availability of large-scale public data and an increase in computing power. DNNs have led to breakthroughs in a wide range of research areas such as computer vision (CV), natural language processing (NLP), speech recognition, and autonomous vehicles, to mention just a few. Thanks to the evolution of new learning methodologies and architectures, DNNs often surpass other machine learning methods and sometimes even exceed human performance on an increasing number of tasks (Goodfellow et al., 2016). However, recent DL architectures can be very large, with up to hundreds of layers and millions or even billions of model parameters that are impossible to fit into commodity graphics processing units (GPUs), that often today have between 11 and 24 GB of memory. For example, Brown et al. (2020) introduced an auto-regressive language model, Generative Pre-trained Transformer 3

(GPT-3), that has 175 billion parameters and requires 700 GB of GPU memory in total with the purpose of producing realistic text outputs. In recent work, Du et al. (2022) presented the Generalist Language Model (GLaM), the largest of which has about 1.2 trillion parameters, occupying around 4.9 TB of memory. These are the extremes, but standard convolutional neural network (CNN) models also require large amounts of memory for commodity GPUs. For instance, the ResNet152 by He et al. (2016) and the VGG-16 by Simonyan & Zisserman (2015) have about 60 and 138 million parameters, respectively, and require 24 and 8 GB of memory (with a mini-batch size of 64 and an input resolution of $256 \times 256$). Another example is the recent Vision Transformer (ViT) model, trained by Zhai et al. (2022), that has two billion parameters and occupies 8 GB of memory (only for the parameters); it achieved a new top-1 accuracy of 90.45% on the ImageNet dataset (Krizhevsky et al., 2012). A popular and powerful architecture for semantic image segmentation, the two-dimensional (2D) U-Net, proposed by Ronneberger et al. (2015) comprises about 8 million parameters and requires around 18 GB of GPU memory (with a mini-batch size of 64, an input resolution of $256 \times 256$, and 32 base filters).

A drawback of DNNs is that they are often over-parameterized, making them expensive, both computationally and memory-wise, but also making them prone to overfitting. Hence, modern DNNs are often expensive resources-wise, which becomes a problem on platforms with limited hardware, especially on embedded systems but also on regular laptop and desktop computers. Over the years, much effort has been made to reduce the number of floating point operations (FLOPs) and the overall network sizes. For instance, Tan & Le (2019a) presented more efficient network architectures by scaling and identifying the balance between network depth, width, and resolution to obtain better performance. Another direction has been to compress DNNs, often to compress the model weights (Hubara et al., 2017). Most works in model compression aim to reduce the model's weight size, but this is not the most memory-consuming part (Mishra et al., 2020). The most memory-consuming parts of a neural network are the intermediate activation maps (Rhu et al., 2016). This becomes especially apparent in medical imaging, when processing three-dimensional (3D) volume images (Milletari et al., 2016; Kamnitsas et al., 2017). For example, with 32-bit floating-point representations, a single $256 \times 256 \times 128$ image (128 slices, each with resolution $256 \times 256$) occupy about 34 MB, a single $32 \times 256 \times 256 \times 128$ activation map (32 filters for 128 slices of resolution $256 \times 256$) occupy about 1 GB, while their corresponding $32 \times 3 \times 3 \times 3$ weight tensor only occupy about 3 kB.

When training a DNN, there are five categories of memory allocated on the GPU (Gao et al., 2020): model parameters (weights), feature maps, gradient maps (for back-propagation), ephemeral tensor (NVIDIA CUDA Deep Neural Network library (cuDNN) workspace and temporary tensors) and resident buffer (CUDA context, internal tensor fragmentation, and allocator reservation). Rhu et al. (2016) showed that the more layers a network has, the larger the fraction of memory allocated for the feature maps on a GPU. For instance, the percentage is always over 50% for a large model such as *e.g.*, VGG-16 (Simonyan & Zisserman, 2015). It is even higher when the data is very high-dimensional, for example, when training on 3D medical images (Rhu et al., 2016). A potential solution to this problem would be to compress the intermediate feature maps in the network layers. It should be noted that the three terms "activation maps", "feature maps" and "intermediate feature maps" are used interchangeably in the rest of the paper.

In this study, we systematically investigated three approaches to compress the intermediate feature maps of a DNN to decrease the memory footprint on a GPU. The study was performed on natural and medical images for classification and segmentation, respectively. Unlike much previous work, the proposed method generally applies to any DNN model. Our method is uniquely versatile and capable of handling any combination of layers within the model. This adaptability allows us to optimize the compression process based on the specific characteristics and interactions of the layers. Whether it is a single layer, a complex combination of multiple layers, or a whole network, our method consistently delivers efficient and effective compression. This demonstrates the potential of our approach for enhancing the performance and scalability of models across a wide range of applications.

Our main contributions can be summarized as follows:

1. We propose a novel way to compress the intermediate feature maps of a DNN. Our method stands out in its ability to handle any combination of layers within the model.

2. We prove that the proposed method works as a regularizer that acts on the layer weight gradients and, by that, can help reduce overfitting.

3. We demonstrate the utility of the proposed method by comparing the GPU memory usage and performance of baseline models to that of models using the proposed compression method. The comparisons were made with respect to graphical memory use, training time, prediction time, and relevant model evaluation metrics. The experiments were performed on multiple datasets and network architectures.

## 2 Related Work

Many lightweight CNN architectures have emerged in recent years that try to find a compromise between computational complexity and model accuracy (Howard et al., 2017; Iandola et al., 2016; Deng et al., 2020). For instance, Redmon et al. (2016) proposed the You Only Look Once model, that includes *e.g.*, $1 \times 1$ convolutions and depthwise-separable convolutions. Standard compression methods include for instance to prune weights (Yu et al., 2018; Hagiwara, 1993; Lee et al., 2018) or to quantize the weights or activations (Dally, 2015; Rastegari et al., 2016; Gysel et al., 2018). Other common approaches include adjusting pre-trained networks by substituting the convolutional kernels with low-rank factorizations of them and grouped convolutions (Xue et al., 2013; 2014). In particular, in these works, a pre-trained weight matrix is factorized into a product of two smaller matrices, serving the same purpose as the original weight matrix but consuming less memory.

Another approach is knowledge distillation, where a smaller network learns to replicate the function of a more extensive and more complicated network Buciluǎ et al. (2006). Buciluǎ et al. (2006) investigated the possibility of shrinking model sizes by constructing an ensemble of models and then using those models to teach a student network. They trained a student network utilizing supervision from the pseudo labels supplied by the teacher network. This phase was done by employing a teacher network to label a significant amount of previously unlabeled data. They found that the performance was comparable to the first ensemble despite the network being one thousand times smaller. Hinton et al. (2015) introduced a neural network knowledge distillation strategy in which a single model was trained to distil the knowledge of a group of other models.

Different forms of activation regularization have been presented in the literature and include common methods such as dropout (Srivastava et al., 2014), batch normalization (Ioffe & Szegedy, 2015), layer normalization (Ba et al., 2016), and $\ell_2$ regularization (Merity et al., 2017). Additionally, the impact of sparse activations has been investigated, for instance, in sparse autoencoders (Hu et al., 2018) and in CNNs (Glorot et al., 2011) where rectified linear unit (ReLU) activations give rise to sparsity in the activation maps. Glorot et al. (2011) also briefly explored the usage of $\ell_1$ regularization to enhance the regularization effect. On the other hand, the usefulness of the regularizer has not been well investigated. Wang & Cheng (2016) presented a low-rank and group sparse tensor decomposition technique to speed up the CNN test phase. In their approach, the kernel tensor was decomposed for each convolutional layer into the sum of a limited number of low-rank tensors. Back-propagation was then used to fine-tune the whole network to meet its ultimate classification target by replacing all of the original kernels with the estimated tensors.

There are a limited number of works that address activation map compression. Dong et al. (2017) aimed to forecast which output activations are zero to avoid calculating them, hence decreasing the number of multiply-accumulate operations executed. Their strategy also altered the network architecture but did not highlight the sparsity of the activation maps. Gudovskiy et al. (2018) compressed the feature maps by first projecting them to binary vectors and then using a nonlinear dimensionality reduction method. It performed marginally better than quantizing the activation maps. Dhillon et al. (2018) proposed stochastic activation pruning as an adversarial defense. This method prunes a random subset of the activations and rescales the remaining ones to compensate for the pruned activations. However, their proposed sampling method yields the highest performance when 100% of the samples are selected, resulting in no change in sparsity. In other works, Alwani et al. (2016) and Chen et al. (2016) reduce the network's memory consumption by recomputing the activation maps rather than saving them, which is effective but incurs a heavy computational cost.

Rhu et al. (2018) explored three techniques for lossless activation map compression: run-length encoding (Robinson & Cherry, 1967), zero-value compression, and Zlib compression (Deutsch & Gailly, 1996). Because of high computational costs, Zlib cannot be used in practice. The other two are hardware-friendly and achieve competitive compression when sparsity is high. Huffman coding (Huffman, 1952) and arithmetic coding (Rissanen & Langdon, 1979) are two other methods proposed in the research community for accomplishing lossless weight compression. Finder et al. (2022) proposed Wavelet Compressed Convolution (WCC), a compression method based on the Wavelet Transform (WT) for high-resolution feature maps. The WCC was integrated with point-wise convolutions and light quantization (1–4 bits).

We have developed a novel compression method that acts on the activation maps (primarily in CNNs). The activation maps are decompressed in the backward pass and used instead of the original activation maps when computing the gradient of the loss with respect to trainable parameters in that layer. We also show that this activation compression induces a regularization on the compressed layer's weight gradients, similar to a gradient penalty (Gulrajani et al., 2017).

# 3 Proposed Methods

We propose to compress the intermediate activation maps in the layers of a neural network to reduce the GPU memory use. The compressed feature maps are kept in GPU memory instead of the original feature maps until they are decompressed (or reconstructed) when needed in the back-propagation pass when computing the gradient of the loss function with respect to the network parameters in that layer.

We evaluate three different compression techniques for the activation maps: WT (Haar, 1909; Zweig, 1976; Mallat, 1989), Discrete Cosine Transform (DCT) (Ahmed et al., 1974), and what we denote as Simple Thresholding (ST). The DCT is used in JPEG[1] and MPEG-C[2], while the WT is employed in JPEG 2000[3]. In the WT method, the feature maps are first WT transformed, and the WT coefficients are then thresholded. In the DCT method, we apply DCT to image blocks, using standard DCT $16 \times 16$ blocks. Next, each block is thresholded. The thresholded feature maps are reconstructed by computing the inverse transforms in both the WT and DCT methods. We further evaluate the ST, in which thresholding is performed directly on the feature maps. The thresholded feature maps are converted to sparse tensors that are stored on the GPU. In the following section, we describe these compression techniques in more detail.

## 3.1 Compression Methods

### 3.1.1 The Wavelet Transform

This work employs the WT (Haar, 1909; Zweig, 1976; Mallat, 1989) to compress 2D and 3D feature maps at multiple resolutions. Let $\boldsymbol{X} \in \mathbb{R}^{2m \times 2m}$ be a real array, and let $\boldsymbol{g}$ and $\boldsymbol{h}$ be low-pass and high-pass filters related to each other by what's called a mother wavelet.

The decomposition of $\boldsymbol{X}$ along the $x$-direction in the first level results in,

$$\boldsymbol{Y}_{\text{low}}^{(1)} = (\boldsymbol{X} * \boldsymbol{g}) \downarrow 2, \tag{1}$$

$$\boldsymbol{Y}_{\text{high}}^{(1)} = (\boldsymbol{X} * \boldsymbol{h}) \downarrow 2, \tag{2}$$

where $\boldsymbol{Y}_{\text{low}}^{(1)} \in \mathbb{R}^{m \times 2m}$ and $\boldsymbol{Y}_{\text{high}}^{(1)} \in \mathbb{R}^{m \times 2m}$ are wavelet coefficients with respect to low and high pass filters along $x$-direction. The $\downarrow$ is a subsampling operation. Equation 1 and 2 can be extended along $y$-direction to

---

[1] https://en.wikipedia.org/wiki/JPEG
[2] https://en.wikipedia.org/wiki/Moving_Picture_Experts_Group
[3] https://en.wikipedia.org/wiki/JPEG_2000

accomplish a 2D wavelet decomposition at one level,

$$\boldsymbol{Y}_{\text{low,low}}^{(1)} = (\boldsymbol{Y}_{\text{low}}^{(1)} * \boldsymbol{g}^{\top}) \downarrow 2, \tag{3}$$

$$\boldsymbol{Y}_{\text{low,high}}^{(1)} = (\boldsymbol{Y}_{\text{low}}^{(1)} * \boldsymbol{h}^{\top}) \downarrow 2, \tag{4}$$

$$\boldsymbol{Y}_{\text{high,low}}^{(1)} = (\boldsymbol{Y}_{\text{high}}^{(1)} * \boldsymbol{g}^{\top}) \downarrow 2, \tag{5}$$

$$\boldsymbol{Y}_{\text{high,high}}^{(1)} = (\boldsymbol{Y}_{\text{high}}^{(1)} * \boldsymbol{h}^{\top}) \downarrow 2, \tag{6}$$

where $\boldsymbol{Y}_{\text{low,low}}^{(1)}$, $\boldsymbol{Y}_{\text{low,high}}^{(1)}$, $\boldsymbol{Y}_{\text{high,low}}^{(1)}$ and $\boldsymbol{Y}_{\text{high,high}}^{(1)} \in \mathbb{R}^{m \times m}$ denote wavelet coefficients in the first level of the wavelet decomposition. The process is repeated recursively at higher levels using the low-pass filtered outputs. Figure 1 illustrates a two-level wavelet decomposition. Thresholding is applied to all components, *i.e.* to $\boldsymbol{Y}_{\text{low,high}}^{(1)}$, $\boldsymbol{Y}_{\text{high,low}}^{(1)}$, $\boldsymbol{Y}_{\text{high,high}}^{(1)}$, $\boldsymbol{Y}_{\text{low,high}}^{(2)}$, $\boldsymbol{Y}_{\text{high,low}}^{(2)}$, $\boldsymbol{Y}_{\text{high,high}}^{(2)}$, and $\boldsymbol{Y}_{\text{low,low}}^{(2)}$.

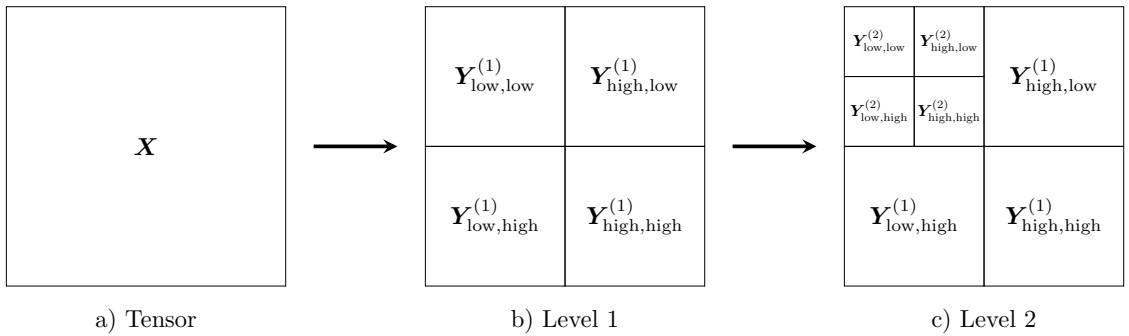

a) Tensor      b) Level 1      c) Level 2

Figure 1: Illustration of a two-level WT decomposition. After decomposition, the top left component $\boldsymbol{Y}_{\text{low,low}}^{(1)}$ in (b) contains the low-frequency component of the image in (a), and similarly, the top-left corner in (c) contain the low-frequency components of the top-left down-scaled part in (b). Thresholding is then applied to all components, such that the smallest wavelet coefficients are set to zero, giving minimal losses when the image is reconstructed later on.

Assuming we employ only one-level WT decomposition, using Equation 1–6, the $\boldsymbol{X}$ can be reconstructed as,

$$\boldsymbol{X} = (\boldsymbol{Y}_{\text{low,low}}^{(1)} *^{\top} \boldsymbol{g}^{\top}) *^{\top} \boldsymbol{g} + (\boldsymbol{Y}_{\text{low,high}}^{(1)} *^{\top} \boldsymbol{h}^{\top}) *^{\top} \boldsymbol{g}$$
$$+ (\boldsymbol{Y}_{\text{high,low}}^{(1)} *^{\top} \boldsymbol{g}^{\top}) *^{\top} \boldsymbol{h} + (\boldsymbol{Y}_{\text{high,high}}^{(1)} *^{\top} \boldsymbol{h}^{\top}) *^{\top} \boldsymbol{h},$$
$$:= \boldsymbol{X}_{\text{low,low}} + \boldsymbol{X}_{\text{low,high}} + \boldsymbol{X}_{\text{high,low}} + \boldsymbol{X}_{\text{high,high}}, \tag{7}$$

where $*^{\top}$ denotes a transposed convolution. For higher decomposition levels, $\boldsymbol{X}$ still has the same form as in Equation 7. It should be noted that we used a three-levels WT decomposition for all experiments in this work.

The Wavelet transform can also be expressed more compactly as,

$$\boldsymbol{Y} = \boldsymbol{W}\boldsymbol{X}, \tag{8}$$

where $\boldsymbol{W}$ is an orthogonal matrix that performs the WT.

### 3.1.2 The Discrete Cosine Transform

In the DCT method, we first split the matrix into $16 \times 16$ blocks. We then employ the DCT of type II (DCT-II) proposed in Ahmed et al. (1974) to compress each block. We used an invertible DCT formulation in this work.

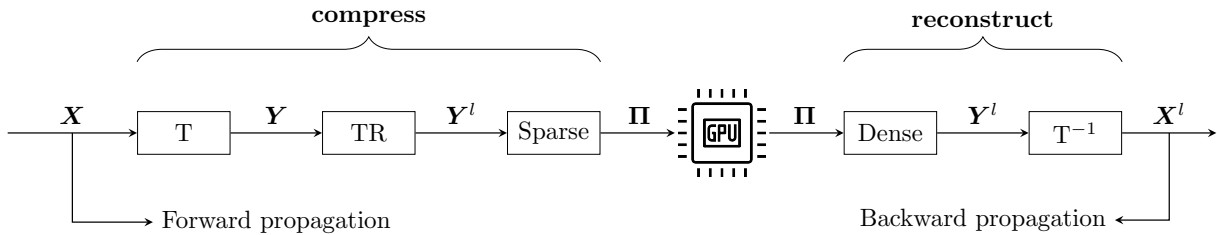

Figure 2: The proposed compression method. The transform T and $T^{-1}$ denote decomposition and reconstruction operation, respectively. The TR denotes the thresholding step, while Sparse and Dense stand for sparse encoding and dense decoding, respectively. First, the input $X$ is: (i) decomposed into multi-level coefficients $Y$ if the WT is used, or (ii) split in $16 \times 16$ blocks and applied DCT for each block, if the DCT is used, or (iii) unchanged, if the ST is used (see Section 3.1). Second, each element in $Y$ is thresholded to produce a thresholded tensor $Y^l$ (Section 3.2). Third, the $Y^l$ is encoded in a sparse format, denoted $\Pi$, to decrease the GPU memory usage, and these sparse tensors are then stored on the GPU instead of storing $X$ there (Section 3.3). Fourth, $Y^l$ can be retrieved by reconstructing the sparse tensor $\Pi$. Last, the thresholded version of $X$, the $X^l$, is used to compute the gradient instead of $X$ in the backward propagation (Section 3.1). In summary, we propose to save GPU memory by storing $\Pi$ instead of $X$ in the forward propagation. In the backward propagation, $X^l$, a decompressed version of $X$, is used when computing the gradients.

Let $X \in \mathbb{R}^{2m \times 2m}$ be a real array, which is a split block. In this case, $m$ is equivalent to 8. The DCT of $X$ is defined as,

$$y_{i,j} = \frac{1}{m} \sum_{u=0}^{2m-1} \sum_{v=0}^{2m-1} \Lambda_i \Lambda_j \cdot \cos \frac{(2u+1)\pi i}{4m} \cdot \cos \frac{(2v+1)\pi j}{4m} \cdot x_{u,v},$$

with $x_{u,v} \in X$ and $y_{i,j} \in Y$. The $u$ and $i$ denote the row index of $X$ and $Y$, respectively. The $v$ and $j$ denote the column index of $X$ and $Y$, respectively. The $\Lambda_k$ is

$$\Lambda_k = \begin{cases} \frac{1}{\sqrt{2}}, & \text{if } k = 0, \\ 1, & \text{otherwise.} \end{cases}$$

The corresponding inverse 2D DCT is defined,

$$x_{u,v} = \frac{1}{m} \sum_{i=0}^{2m-1} \sum_{j=0}^{2m-1} \Lambda_i \Lambda_j \cdot \cos \frac{(2u+1)\pi i}{4m} \cdot \cos \frac{(2v+1)\pi j}{4m} \cdot y_{i,j},$$

with $x_{u,v} \in X$ and $y_{i,j} \in Y$.

The DCT is also a linear function, and hence, similar to the WT, we can also express it more compactly as,

$$Y = BX, \tag{9}$$

with $B$ the (orthogonal) DCT.

### 3.1.3 The Simple Thresholding

With the ST method, the activation maps are not transformed but are directly thresholded. The thresholding method we employed is described below.

## 3.2 Thresholding

The second step of the compression is to threshold the (transformed, in the case of WT and DCT) activation maps. The core idea is to determine a threshold, where values below the threshold are set to zero and thus

removed. The threshold is found using a "top-$k$ approach" that aims to retain a certain number of elements. It is detailed below.

Let $\boldsymbol{Y}$ be a real array of arbitrary dimension, and let

$$\boldsymbol{y} := \text{vec}(\boldsymbol{Y}) = (y_1, y_2, \ldots, y_N)^\top$$

be a vectorization of $\boldsymbol{Y}$, where $N \geq 1$ is the number of elements in $\boldsymbol{y}$. Without loss of generality, we assume the elements are sorted in a monotonically decreasing order of their absolute values.

The index, $k$, of a threshold element, $y_k$, of $\boldsymbol{y}$ is determined as,

$$k = \left\lfloor \frac{1}{\lambda} N \right\rfloor$$

where $1 \leq \lambda < \infty$ is the compression ratio and $\lfloor \cdot \rfloor$ is the floor function. The compression ratio is defined as the ratio between the uncompressed size (*e.g.*, an array has $N$ elements) and compressed size (*e.g.*, a compressed array has $k$ elements). The impact of $\lambda$ on three arrays with different characteristics is illustrated in Figure 3. The first array has a substantial number of low values, the second has a normal-like distribution, and the third has a substantial number of high values.

We construct binary arrays, $\boldsymbol{M}^l$ and $\boldsymbol{M}^s$, that, when vectorized, have elements

$$m_i^l = \begin{cases} 1, & \text{if } i \leq k, \\ 0, & \text{otherwise,} \end{cases}$$

and

$$m_i^s = 1 - m_i^l = \begin{cases} 1, & \text{if } i > k, \\ 0, & \text{otherwise,} \end{cases}$$

respectively (large and small), giving masks for large and small values as,

$$\boldsymbol{M}^l = \text{vec}^{-1}(\boldsymbol{m}^l) \quad \text{and} \quad \boldsymbol{M}^s = \text{vec}^{-1}(\boldsymbol{m}^s).$$

Hence, we determine the $k$ largest absolute values of the given array. We denote the special case of no compression when $\lambda = 1$ as the baseline (BL).

We then select the large and small values as,

$$\boldsymbol{Y}^l := \boldsymbol{M}^l \odot \boldsymbol{Y} \quad \text{and} \quad \boldsymbol{Y}^s := \boldsymbol{M}^s \odot \boldsymbol{Y} = (\boldsymbol{1} - \boldsymbol{M}^l) \odot \boldsymbol{Y}, \tag{10}$$

respectively, and decompose the $\boldsymbol{Y}$ by

$$\boldsymbol{Y} = (\boldsymbol{M}^l + \boldsymbol{M}^s) \odot \boldsymbol{Y} = \boldsymbol{M}^l \odot \boldsymbol{Y} + \boldsymbol{M}^s \odot \boldsymbol{Y} = \boldsymbol{Y}^l + \boldsymbol{Y}^s, \tag{11}$$

where compression is achieved by removing the smallest (in absolute value) $N - k$ elements of $\boldsymbol{Y}$, *i.e.* setting $\boldsymbol{Y}^s$ to zero.

### 3.3 Sparse Tensor

In the third step, the thresholded coefficients are stored in a sparse tensor format (the coordinate (COO) format) to actually reduce the memory consumption after thresholding. Hence, the sparse tensor is what is actually stored on the GPU. The sparse representation is computed and denoted as,

$$\boldsymbol{\Pi} = \text{Sparse}(\boldsymbol{Y}^l), \tag{12}$$

and when retrieved in the back-propagation phase, the reconstruction of the dense tensor is denoted,

$$\boldsymbol{Y}^l = \text{Dense}(\boldsymbol{\Pi}). \tag{13}$$

See Figure 2 for the entire process of compressing a tensor (transforming and thresholding it), storing it in sparse format on GPU, and retrieving and reconstructing it.

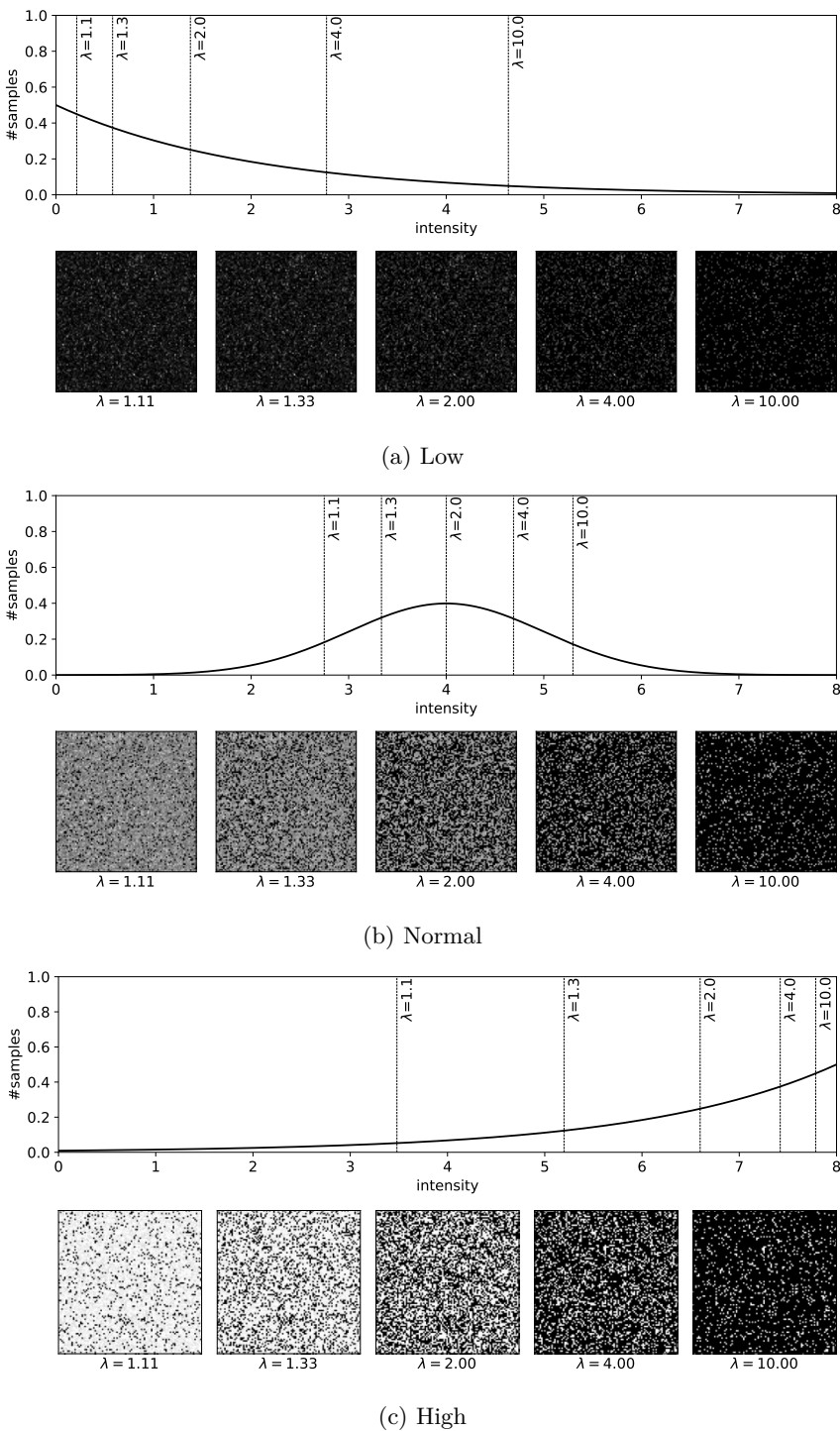

Figure 3: Effects of compression ratio, $\lambda$, on three arrays with: **(a)** substantial number of low values, **(b)** normal-like distribution and **(c)** substantial number of high values.

# 4 Analysis of the Regularizing Effect

We show in this section that the proposed compression methods, using any of the proposed compression methods, induce a regularization effect that acts on a compressed layer's weight gradients. We show it in the general case and then demonstrate it also for three common layers: convolution, ReLU, and fully connected layers.

## 4.1 Forward and Backward Propagation

Suppose that we have a feed-forward neural network (NN) with $n$ layers,

$$\widehat{\boldsymbol{V}} = f(\boldsymbol{U}; \boldsymbol{\theta}) = f^{(n)}(f^{(n-1)}(\cdots f^{(2)}(f^{(1)}(\boldsymbol{U}))\cdots)), \tag{14}$$

where $\boldsymbol{U}$ and $\boldsymbol{V}$ are real input and output arrays of the network, and $\boldsymbol{\theta} = (\boldsymbol{\omega}^{(1)}, \ldots, \boldsymbol{\omega}^{(n)})$ denotes all trainable parameters for all layers. The $f^{(1)}$ is the first layer of the network, $f^{(2)}$ is the second layer, and so on.

When we apply the network to an input, $\boldsymbol{U}$, the data flow forward through the network. The input, $\boldsymbol{U}$, provides the initial information, which is then propagated to all hidden units before producing an output prediction, $\widehat{\boldsymbol{V}}$. Together with a ground-truth target, $\boldsymbol{V}$, we want to find network parameters that minimize a scalar loss function, $L(\boldsymbol{V}, \widehat{\boldsymbol{V}})$. When using first-order optimization algorithms to update the network's parameters, the back-propagation algorithm (Rumelhart et al., 1986) is the most common way to determine the gradient of the loss, $\nabla_{\boldsymbol{\theta}} L(\boldsymbol{V}, \widehat{\boldsymbol{V}})$, with respect to the model parameters, $\boldsymbol{\theta}$.

Consider the $m$-th layer of the network in Equation 14. Let the input and output of the $m$-th layer be $\boldsymbol{X}$ and $\boldsymbol{Z}$, *i.e.*,

$$\boldsymbol{Z} = f^{(m)}(\boldsymbol{X}), \tag{15}$$

where the $f^{(m)}$ would typically either be an affine function or an element-wise non-linear activation function. We denote by $\nabla_{\boldsymbol{X}} L$ and $\nabla_{\boldsymbol{Z}} L$ the gradients of the loss, $L(\boldsymbol{V}, \widehat{\boldsymbol{V}})$, with respect to $\boldsymbol{X}$ and $\boldsymbol{Z}$, respectively, and $\nabla_{\boldsymbol{\omega}^{(m)}} L$ denote the gradient of the loss with respect to the $m$-th layer's trainable parameters, $\boldsymbol{\omega}^{(m)}$. We further denote by $\nabla^l_{\boldsymbol{\omega}^{(m)}} L$ the gradient of the loss, $L$, with respect to the layer's weights, $\boldsymbol{\omega}^{(m)}$, when the input $\boldsymbol{X}$ is replaced by the reconstructed $\boldsymbol{X}^l$.

The claims below are made under the following assumptions.

**Assumption 1.** *The elements of the binary array, $\boldsymbol{M}^l$, in Equation 10, independently follow a Bernoulli distribution with probability $\frac{1}{\lambda}$.*

**Assumption 2.** *The input $\boldsymbol{X}^l$ to a compressed layer is in one of the following forms, either*

$$\boldsymbol{X}^l = \boldsymbol{D}_l\big(\boldsymbol{M}^l \odot (\boldsymbol{C}_l \boldsymbol{X})\big) \quad or \quad \boldsymbol{X}^l = \big(\boldsymbol{M}^l \odot (\boldsymbol{X} \boldsymbol{C}_r)\big)\boldsymbol{D}_r,$$

*with the $\boldsymbol{C}_t$ and $\boldsymbol{D}_t$, for $t \in \{r, l\}$, being arbitrary real (or complex) arrays, and with $\boldsymbol{M}^l$ a binary array as in Equation 10.*

The following lemma allows us to determine a regularization effect on the layer weight gradient when the layer is compressed, *i.e.*, when a layer's input, $\boldsymbol{X}$, is replaced in the backward pass of back-propagation by its (possibly transformed) thresholded (and possibly back-transformed) version, $\boldsymbol{X}^l$.

**Lemma 1.** *Under Assumption 1 and Assumption 2, if $\nabla_{\boldsymbol{\omega}^{(m)}} L$ either has the form,*

$$\nabla_{\boldsymbol{\omega}^{(m)}} L = \boldsymbol{X}^\top g(\nabla_{\boldsymbol{Z}} L), \tag{16}$$

*or the form,*

$$\nabla_{\boldsymbol{\omega}^{(m)}} L = \boldsymbol{X} \, g(\nabla_{\boldsymbol{Z}} L), \tag{17}$$

*where $g$ is a linear function, then*

$$\mathbb{E}_{\boldsymbol{M}^l}\left[\|\nabla^l_{\boldsymbol{\omega}^{(m)}} L\|_F^2\right] \leq \frac{1}{\lambda} \cdot \|\boldsymbol{C}_t \boldsymbol{X}\|_F^2 \cdot \begin{cases} \|\boldsymbol{D}_t^\top g(\nabla_{\boldsymbol{Z}} L)\|_F^2, & if \ t = l, \\ \|\boldsymbol{D}_t \, g(\nabla_{\boldsymbol{Z}} L)\|_F^2, & if \ t = r, \end{cases}$$

where $1 \leq \lambda < \infty$ is the compression ratio, $t \in \{r, l\}$ denotes the left or right versions of the compression in Assumption 2, and $\|\cdot\|_F$ denotes the Frobenius norm. The expectation is taken over all possible binary matrices, $\boldsymbol{M}^l$.

*Proof.* To simplify the notation, let

$$\boldsymbol{A} = g(\nabla_{\boldsymbol{Z}} L),$$

and let

$$\boldsymbol{E} = \boldsymbol{C}_l \boldsymbol{X},$$

and expand Equation 16, we see that,

$$
\begin{aligned}
\mathbb{E}_{\boldsymbol{M}^l}\left[\|\nabla^l_{\boldsymbol{\omega}^{(m)}} L\|_F^2\right] &= \mathbb{E}_{\boldsymbol{M}^l}\left[\|(\boldsymbol{X}^l)^\top \boldsymbol{A}\|_F^2\right] \\
&= \mathbb{E}_{\boldsymbol{M}^l}\left[\|(\boldsymbol{M}^l \odot \boldsymbol{E})^\top \boldsymbol{D}_l^\top \boldsymbol{A}\|_F^2\right] \\
&\leq \mathbb{E}_{\boldsymbol{M}^l}\left[\|\boldsymbol{M}^l \odot \boldsymbol{E}\|_F^2\right] \cdot \|\boldsymbol{D}_l^\top \boldsymbol{A}\|_F^2 \\
&= \mathbb{E}_{\boldsymbol{M}^l}\left[\sum_{m_i^l \in \boldsymbol{M}^l, e_i \in \boldsymbol{E}} (m_i^l e_i)^2\right] \cdot \|\boldsymbol{D}_l^\top \boldsymbol{A}\|_F^2 \\
&= \sum_{m_i^l \in \boldsymbol{M}^l, e_i \in \boldsymbol{E}} \mathbb{E}\left[(m_i^l)^2\right] \cdot e_i^2 \cdot \|\boldsymbol{D}_l^\top \boldsymbol{A}\|_F^2 \\
&= \sum_{e_i \in \boldsymbol{E}} \frac{1}{\lambda} \cdot e_i^2 \cdot \|\boldsymbol{D}_l^\top \boldsymbol{A}\|_F^2 \\
&= \frac{1}{\lambda} \cdot \sum_{e_i \in \boldsymbol{E}} e_i^2 \cdot \|\boldsymbol{D}_l^\top \boldsymbol{A}\|_F^2 \\
&= \frac{1}{\lambda} \cdot \|\boldsymbol{E}\|_F^2 \cdot \|\boldsymbol{D}_l^\top \boldsymbol{A}\|_F^2, \\
&= \frac{1}{\lambda} \cdot \|\boldsymbol{C}_l \boldsymbol{X}\|_F^2 \cdot \|\boldsymbol{D}_l^\top \boldsymbol{A}\|_F^2,
\end{aligned}
$$

where $a_i \in \boldsymbol{A}$ denotes an element (the $i$-th element in any order) from an arbitrary shaped array $\boldsymbol{A}$. The inequality is the Cauchy-Schwarz, and we use that the expectation of a squared Bernoulli random variable has the same expectation as the Bernoulli random variable itself.

Analogously, we expand Equation 17 and have that

$$
\begin{aligned}
\mathbb{E}_{\boldsymbol{M}^l}\left[\|\nabla^l_{\boldsymbol{\omega}^{(m)}} L\|_F^2\right] &= \mathbb{E}_{\boldsymbol{M}^l}\left[\|(\boldsymbol{M}^l \odot (\boldsymbol{X}\boldsymbol{C}_r))\boldsymbol{D}_r \boldsymbol{A}\|_F^2\right] \\
&\leq \mathbb{E}_{\boldsymbol{M}^l}\left[\|\boldsymbol{M}^l \odot (\boldsymbol{X}\boldsymbol{C}_r)\|_F^2\right] \cdot \|\boldsymbol{D}_r \boldsymbol{A}\|_F^2 \\
&= \frac{1}{\lambda} \cdot \|\boldsymbol{X}\boldsymbol{C}_r\|_F^2 \cdot \|\boldsymbol{D}_r \boldsymbol{A}\|_F^2
\end{aligned}
$$

where we followed the same steps as in the expansion of Equation 16 above. $\qquad\square$

Lemma 1 tells us that by adjusting the compression ratio, $\lambda$, we control a bound on $\mathbb{E}_{\boldsymbol{M}^l}\left[\|\nabla^l_{\boldsymbol{\omega}^{(m)}} L\|_2^2\right]$. When $\lambda$ is set to 1, then there is no compression, and the layer is thus unaffected. When $\lambda$ is increased towards $\infty$ (*i.e.* $k$ decreases), the $\nabla^l_{\boldsymbol{\omega}^{(m)}} L$ is shrunk towards zero (its expected squared Frobenius norm goes to zero).

We now use Lemma 1 to show that all the proposed compression methods induce a gradient regularization on the layer's weight gradients.

**Theorem 1.** *Under Assumption 1 and Assumption 2, if $\nabla_{\boldsymbol{\omega}^{(m)}} L$ either has the form,*

$$\nabla_{\boldsymbol{\omega}^{(m)}} L = \boldsymbol{X}^\top g(\nabla_{\boldsymbol{Z}} L), \tag{18}$$

*or the form,*

$$\nabla_{\boldsymbol{\omega}^{(m)}} L = \boldsymbol{X} \, g(\nabla_{\boldsymbol{Z}} L), \tag{19}$$

*where g is a linear function, then*

$$\mathbb{E}_{\boldsymbol{M}^l}\left[\|\nabla_{\boldsymbol{\omega}^{(m)}}^l L\|_F^2\right] \le \frac{1}{\lambda} \cdot \begin{cases} \|\boldsymbol{X}\|_F^2 \cdot \|g(\nabla_{\boldsymbol{Z}} L)\|_F^2, & \text{for ST,} \\ \|\boldsymbol{W}\boldsymbol{X}\|_F^2 \cdot \|\boldsymbol{W}^{-\top} g(\nabla_{\boldsymbol{Z}} L)\|_F^2, & \text{for WT,} \\ \|\boldsymbol{B}\boldsymbol{X}\|_F^2 \cdot \|\boldsymbol{B}^{-\top} g(\nabla_{\boldsymbol{Z}} L)\|_F^2, & \text{for DCT,} \end{cases} \tag{20}$$

*where $\boldsymbol{W}$ is the WT and $\boldsymbol{B}$ is the (orthogonal) DCT.*

*Proof.* We consider three cases, corresponding to the three proposed compression methods.

*Case 1: The ST.* We let $\boldsymbol{C}_t = \boldsymbol{D}_t = \boldsymbol{I}$ be identity matrices, for either value of $t \in \{l, r\}$, then invoke Lemma 1.

*Case 2: The WT.* The WT is an orthogonal linear transform, and we can express the layer's inputs in either of the two forms, as

$$\boldsymbol{X}^l = \boldsymbol{W}_l^{-1}\big(\boldsymbol{M}^l \odot (\boldsymbol{W}_l \boldsymbol{X})\big) = \big(\boldsymbol{M}^l \odot (\boldsymbol{X}\boldsymbol{W}_r)\big)\boldsymbol{W}_r^{-1}, \tag{21}$$

and invoke Lemma 1.

*Case 3: The DCT.* For simplicity, we only consider the orthogonal discrete cosine transform, but the following also holds for a non-orthogonal DCT, just with different forward and backward transform matrices. We write the inputs to the layer as,

$$\boldsymbol{X}^l = \boldsymbol{B}^{-1}\big(\boldsymbol{M}^l \odot (\boldsymbol{B}\boldsymbol{X})\big) \tag{22}$$

where $\boldsymbol{B}$ is the discrete cosine transform, and then we invoke Lemma 1. □

Before showing that this also applies to the result of a convolution, we first introduce a matrix representation of a convolution (see, *e.g.*, Goodfellow et al., 2016). Consider two real arrays, $\boldsymbol{A}$ and $\boldsymbol{B}$, of arbitrary dimension, and their discrete convolution, which we write as

$$\begin{aligned} \boldsymbol{A} * \boldsymbol{B} &= \text{vec}^{-1}\left(\text{vec}(\boldsymbol{A}) \cdot \boldsymbol{B}_r\right) \\ &= \text{vec}^{-1}\left(\boldsymbol{A}_r \cdot \text{vec}(\boldsymbol{B})\right), \end{aligned}$$

where $*$ denote the (discrete) convolution operator, and $\boldsymbol{A}_r$ and $\boldsymbol{B}_r$ are the matrices corresponding to the convolution (they are Toeplitz matrices) such that the conversion from convolution to multiplication operation is valid.

We note that,

$$\begin{aligned} \|\boldsymbol{A} * \boldsymbol{B}\|_F^2 &= \|\text{vec}^{-1}\left(\text{vec}(\boldsymbol{A}) \cdot \boldsymbol{B}_r\right)\|_F^2 \\ &= \|\text{vec}(\boldsymbol{A}) \cdot \boldsymbol{B}_r\|_2^2 \\ &\le \|\text{vec}(\boldsymbol{A})\|_2^2 \cdot \|\boldsymbol{B}_r\|_F^2 \\ &= \|\boldsymbol{A}\|_F^2 \cdot \|\boldsymbol{B}_r\|_F^2. \end{aligned} \tag{23}$$

Lemma 1 can also be immediately applied when the gradient, $\nabla_{\boldsymbol{\omega}^{(m)}} L$, is the result of a convolution.

**Corollary 1.** *Under Assumption 1 and Assumption 2, if $\nabla_{\boldsymbol{\omega}^{(m)}} L$ has the form,*

$$\nabla_{\boldsymbol{\omega}^{(m)}} L = \boldsymbol{X} * \boldsymbol{A}, \tag{24}$$

*where $*$ denotes a discrete convolution and $\boldsymbol{A} = g(\nabla_{\boldsymbol{Z}} L)$ with $g$ a linear function, then*

$$\mathbb{E}_{\boldsymbol{M}^l}\left[\|\nabla_{\boldsymbol{\omega}^{(m)}}^l L\|_F^2\right] \le \frac{1}{\lambda} \cdot \begin{cases} \|\boldsymbol{X}\|_F^2 \cdot \|\boldsymbol{A}_r\|_F^2, & \text{for ST,} \\ \|\boldsymbol{W}\boldsymbol{X}\|_F^2 \cdot \|\boldsymbol{W}^{-\top}\|_F^2 \cdot \|\boldsymbol{A}_r\|_F^2, & \text{for WT,} \\ \|\boldsymbol{B}\boldsymbol{X}\|_F^2 \cdot \|\boldsymbol{B}^{-\top}\|_F^2 \cdot \|\boldsymbol{A}_r\|_F^2 & \text{for DCT,} \end{cases}$$

*where $\boldsymbol{W}$ is the WT and $\boldsymbol{B}$ is the (orthogonal) DCT. The $\boldsymbol{A}_r$ is a matrix representation of the convolution with $\boldsymbol{A}$.*

*Proof.* We again consider three cases, corresponding to the three proposed compression methods.

*Case 1: The ST.* We express

$$\boldsymbol{X} * \boldsymbol{A} = \mathrm{vec}^{-1}(\mathrm{vec}(\boldsymbol{X}) \cdot \boldsymbol{A}_r),$$

then

$$
\begin{aligned}
\mathbb{E}_{\boldsymbol{M}^l}\left[\|\nabla^l_{\boldsymbol{\omega}^{(m)}} L\|^2_F\right] &= \mathbb{E}_{\boldsymbol{M}^l}\left[\|\boldsymbol{X}^l * \boldsymbol{A}\|^2_F\right] \\
&= \mathbb{E}_{\boldsymbol{M}^l}\left[\|\mathrm{vec}^{-1}(\mathrm{vec}(\boldsymbol{X}^l) \cdot \boldsymbol{A}_r)\|^2_F\right] \\
&\leq \mathbb{E}_{\boldsymbol{M}^l}\left[\|\boldsymbol{X}^l\|^2_F\right] \cdot \|\boldsymbol{A}_r\|^2_F \\
&= \mathbb{E}_{\boldsymbol{M}^l}\left[\|\boldsymbol{M}^l \odot \boldsymbol{X}\|^2_F\right] \cdot \|\boldsymbol{A}_r\|^2_F.
\end{aligned}
$$

Hence, following Lemma 1, we obtain,

$$\mathbb{E}_{\boldsymbol{M}^l}\left[\|\nabla^l_{\boldsymbol{\omega}^{(m)}} L\|^2_F\right] \leq \frac{1}{\lambda} \cdot \|\boldsymbol{X}\|^2_F \cdot \|\boldsymbol{A}_r\|^2_F.$$

*Case 2: The WT.* We express $\boldsymbol{X}^l$ as in Equation 21, follow the same steps as in the expansion in the ST, apply the Cauchy-Schwarz inequality, and use the same approach as in Lemma 1.

*Case 3: The DCT.* We express $\boldsymbol{X}^l$ as in Equation 22, follow the same steps as in the expansion in the ST, apply the Cauchy-Schwarz inequality, and use the same approach as in Lemma 1. $\square$

## 4.2 Convolution layer

Consider the $m$-th layer of a neural network with input $\boldsymbol{X}$ and output $\boldsymbol{Z}$ as per Equation 15. Suppose that the $m$-th layer in Equation 15 is a convolution layer with weights $\boldsymbol{\Omega}$ and biases $\boldsymbol{b}$, such that,

$$\boldsymbol{Z}_j = f^{(m)}(\boldsymbol{X}) = \boldsymbol{X} * \boldsymbol{\Omega}_j + b_j, \quad \forall j \tag{25}$$

where $\boldsymbol{\Omega}_j$ denotes the $j$-th convolution filter and $b_j$ is the $j$-th bias.

Take the derivatives of the loss with respect to $\boldsymbol{X}$, $b_j$, and $\boldsymbol{\Omega}_j$ in Equation 25, using the multivariate chain rule (*e.g.*, Rumelhart et al., 1986), we express the gradients as (Zhang, 2016),

$$
\begin{aligned}
\nabla_{\boldsymbol{X}} L &= \left(\frac{\partial \boldsymbol{Z}_j}{\partial \boldsymbol{X}}\right)^\top \nabla_{\boldsymbol{Z}_j} L = \boldsymbol{\Omega}'_j * \nabla_{\boldsymbol{Z}_j} L, \\
\nabla_{\boldsymbol{\Omega}_j} L &= \left(\frac{\partial \boldsymbol{Z}_j}{\partial \boldsymbol{\Omega}_j}\right)^\top \nabla_{\boldsymbol{Z}_j} L = \boldsymbol{X} * \nabla_{\boldsymbol{Z}_j} L, \\
\nabla_{b_j} L &= \left(\frac{\partial \boldsymbol{Z}_j}{\partial b_j}\right)^\top \nabla_{\boldsymbol{Z}_j} L = \sum_i (\nabla_{\boldsymbol{Z}_j} L)_i,
\end{aligned}
\tag{26}
$$

with Jacobians $\frac{\partial \boldsymbol{Z}_j}{\partial \boldsymbol{X}}$, $\frac{\partial \boldsymbol{Z}_j}{\partial \boldsymbol{\Omega}_j}$ and $\frac{\partial \boldsymbol{Z}_j}{\partial b_j}$, and where $\boldsymbol{\Omega}'_j$ have all dimensions reversed (flipped) compared to $\boldsymbol{\Omega}_j$.

We thus see that neither $\nabla_{b_j} L$ nor $\nabla_{\boldsymbol{X}} L$ depend on $\boldsymbol{X}$ (and thus neither on $\boldsymbol{X}^l$), but that $\nabla_{\boldsymbol{\Omega}_j} L$ does. The following corollary to Corollary 1 is an immediate consequence.

**Corollary 2.** *Under Assumption 1 and Assumption 2, the expected squared norm of the gradient, $\nabla_{\boldsymbol{\Omega}_j} L$, of the loss function with respect to the $j$-th convolution filter, $\boldsymbol{\Omega}_j$, in a convolution layer is bounded as,*

$$\mathbb{E}_{\boldsymbol{M}^l}\left[\left\|\nabla^l_{\boldsymbol{\Omega}_j^{(m)}} L\right\|^2_F\right] \leq \frac{1}{\lambda} \cdot \begin{cases} \|\boldsymbol{X}\|^2_F \cdot \|\boldsymbol{A}_r\|^2_F, & \text{for ST,} \\ \|\boldsymbol{W}\boldsymbol{X}\|^2_F \cdot \|\boldsymbol{W}^{-\top}\|^2_F \cdot \|\boldsymbol{A}_r\|^2_F, & \text{for WT,} \\ \|\boldsymbol{B}\boldsymbol{X}\|^2_F \cdot \|\boldsymbol{B}^{-\top}\|^2_F \cdot \|\boldsymbol{A}_r\|^2_F & \text{for DCT,} \end{cases}$$

*where $\boldsymbol{A} = \nabla_{\boldsymbol{Z}_j} L$ and $\boldsymbol{A}_r$ is a matrix representation of the convolution with $\boldsymbol{A}$. The $\boldsymbol{W}$ is the WT and $\boldsymbol{B}$ is the (orthogonal) DCT.*

*Proof.* Follows from Equation 26 and Corollary 1. $\square$

### 4.3 ReLU layer

Consider the $m$-th layer of the network in Equation 15 with input, $\boldsymbol{X}$, and output, $\boldsymbol{Z}$. Suppose that the $m$-th layer in Equation 15 is a ReLU layer, such that

$$z_i = \begin{cases} 0, & x_i \leq 0 \\ x_i, & \text{otherwise} \end{cases}$$

with $z_i \in \boldsymbol{Z}$ and $x_i \in \boldsymbol{X}$.

Taking the derivative of the loss $L$ with respect to $x_i$, and using the chain rule, we get

$$\frac{\partial L}{\partial x_i} = \begin{cases} 0, & x_i \leq 0 \\ \frac{\partial L}{\partial z_i}, & \text{otherwise} \end{cases}$$

with $\frac{\partial L}{\partial x_i} \in \nabla_{\boldsymbol{X}} L$ and $\frac{\partial L}{\partial z_i} \in \nabla_{\boldsymbol{Z}} L$.

For the ST, we let

$$\boldsymbol{X}^l = \boldsymbol{M}^l \odot \boldsymbol{X}$$

then

$$\frac{\partial L}{\partial x_i^l} = \begin{cases} 0, & m_i^l\, x_i \leq 0 \\ \frac{\partial L}{\partial z_i}, & \text{otherwise} \end{cases}$$

which we collect as

$$\nabla_{\boldsymbol{X}}^l L = \boldsymbol{M}^l \odot \nabla_{\boldsymbol{X}} L. \tag{27}$$

A corollary to Lemma 1 now follows.

**Corollary 3.** *Under Assumption 1 and Assumption 2, when ST compression is used, the expected gradient of the loss function with respect to the input $\boldsymbol{X}$ of a ReLU layer is bounded as,*

$$\mathbb{E}_{\boldsymbol{M}^l} \left[ \|\nabla_{\boldsymbol{X}}^l L\|_F^2 \right] \leq \frac{1}{\lambda} \cdot \|\nabla_{\boldsymbol{X}} L\|_F^2.$$

*Proof.* We use Equation 27 and have

$$\begin{aligned} \mathbb{E}_{\boldsymbol{M}^l} \left[ \|\nabla_{\boldsymbol{X}}^l L\|_F^2 \right] &= \mathbb{E}_{\boldsymbol{M}^l} \left[ \|\nabla_{\boldsymbol{X}^l} L\|_F^2 \right] \\ &= \mathbb{E}_{\boldsymbol{M}^l} \left[ \|\boldsymbol{M}^l \odot \nabla_{\boldsymbol{X}} L\|_F^2 \right]. \end{aligned}$$

From here, we follow Lemma 1, and obtain

$$\mathbb{E}_{\boldsymbol{M}^l} \left[ \|\nabla_{\boldsymbol{X}}^l L\|_F^2 \right] \leq \frac{1}{\lambda} \cdot \|\nabla_{\boldsymbol{X}} L\|_F^2,$$

which concludes the proof. $\qquad\square$

### 4.4 Fully-connected layer

Suppose that the $m$-th layer of the network in Equation 15 is a fully-connected layer with weights $\boldsymbol{\Omega} \in \mathbb{R}^{p \times m}$, biases $\boldsymbol{b} \in \mathbb{R}^m$, inputs $\boldsymbol{X} \in \mathbb{R}^{n \times p}$, and outputs $\boldsymbol{Z} \in \mathbb{R}^{n \times m}$, such that,

$$\boldsymbol{z}_j = f^{(m)}(\boldsymbol{X}) = \boldsymbol{X}\boldsymbol{\omega}_j + b_j,$$

where $\boldsymbol{\omega}_j$ denotes the $j$-th column of $\boldsymbol{\Omega}$, the $b_j$ is the $j$-th bias, and $\boldsymbol{z}_j$ is the $j$-th column of the output, $\boldsymbol{Z}$.

Taking the derivatives of the loss with respect to $\boldsymbol{X}$, $\boldsymbol{\omega}_j$, and $b_j$, and using the chain rule, we obtain,

$$
\begin{aligned}
\nabla_{\boldsymbol{X}} L &= \frac{\partial \boldsymbol{z}_j}{\partial \boldsymbol{X}} \nabla_{\boldsymbol{z}_j} L = (\boldsymbol{\omega}_j^\top \otimes \boldsymbol{I}) \nabla_{\boldsymbol{z}_j} L \\
\nabla_{\boldsymbol{\omega}_j} L &= \frac{\partial \boldsymbol{z}_j}{\partial \boldsymbol{\omega}_j} \nabla_{\boldsymbol{z}_j} L = \boldsymbol{X}^\top \nabla_{\boldsymbol{z}_j} L, \\
\frac{\partial L}{\partial b_j} &= \frac{\partial \boldsymbol{z}_j}{\partial b_j} \nabla_{\boldsymbol{z}_j} L = \nabla_{\boldsymbol{z}_j} L.
\end{aligned}
\tag{28}
$$

with $\otimes$ the Kronecker product and $\boldsymbol{I}$ the identity matrix with dimensionality implied by context.

We see that neither $\nabla_{\boldsymbol{X}} L$ nor $\frac{\partial L}{\partial b_j}$ is affected by the replacement of $\boldsymbol{X}$ by $\boldsymbol{X}^l$, since they do not contain $\boldsymbol{X}$, but we see that $\nabla_{\boldsymbol{\omega}_j} L$ does. A corollary to Theorem 1 then follows.

**Corollary 4.** *Under Assumption 1 and Assumption 2, the expected squared norm of the gradient, $\nabla_{\boldsymbol{\omega}_j} L$, of the loss function with respect to the $j$-th weight vector, $\boldsymbol{\omega}_j$, of a fully-connected layer is bounded as,*

$$
\mathbb{E}_{\boldsymbol{M}^l}\left[\|\nabla^l_{\boldsymbol{\omega}_j^{(m)}} L\|_F^2\right] \leq \frac{1}{\lambda} \cdot \begin{cases} \|\boldsymbol{X}\|_F^2 \cdot \|\nabla_{\boldsymbol{z}_j} L\|_F^2, & \text{for } ST, \\ \|\boldsymbol{W}\boldsymbol{X}\|_F^2 \cdot \|\boldsymbol{W}^{-\top} \nabla_{\boldsymbol{z}_j} L\|_F^2, & \text{for } WT, \\ \|\boldsymbol{B}\boldsymbol{X}\|_F^2 \cdot \|\boldsymbol{B}^{-\top} \nabla_{\boldsymbol{z}_j} L\|_F^2, & \text{for } DCT, \end{cases}
$$

*where $\boldsymbol{W}$ is the WT and $\boldsymbol{B}$ is the (orthogonal) DCT.*

*Proof.* Follows immediately from Equation 28 and Theorem 1. $\qquad\square$

### 4.5 Connection to Robustness

To date, several studies have investigated regularizing the Jacobian matrix of a DNN's output to improve its robustness. For instance, Jakubovitz & Giryes (2018) used the Frobenius norm of the Jacobian matrix of the network output. This regularization was applied after standard training was completed. Their empirical findings indicated that their proposed approach resulted in a more robust model, while minimally affecting the accuracy of the original network. Sokolić et al. (2017) demonstrated that to achieve good generalization with a DNN with arbitrary depth and width, it is imperative that the network's Jacobian matrix has a limited spectral norm within the vicinity of the training data. In another work, Hoffman et al. (2019) improved training stability by Jacobian regularization, and also showed that it increased the classification margins. They reported that with Jacobian regularization, the model became notably more robust, when evaluated against both random and deliberate input perturbations, while simultaneously maintaining the model's generalization on clean data.

Consider an arbitrary DNN. The full Jacobian of the network can be written as,

$$
\boldsymbol{J} = \boldsymbol{A}^\top \boldsymbol{B}^\top \boldsymbol{C}^\top \boldsymbol{v},
$$

where $\boldsymbol{v}$ is the gradient of the network output, $\boldsymbol{C}$ is the Jacobian of all layer after layer B, $\boldsymbol{A}$ is the Jacobian of all layers before layer B, and $\boldsymbol{B}$ denotes the Jacobian in layer B.

The norm of the full Jacobian is then,

$$
\|\boldsymbol{J}\|_F^2 = \|\boldsymbol{A}^\top \boldsymbol{B}^\top \boldsymbol{C}^\top \boldsymbol{v}\|_F^2 \leq \|\boldsymbol{A}\|_F^2 \|\boldsymbol{B}\|_F^2 \|\boldsymbol{C}\|_F^2 \|\boldsymbol{v}\|_F^2,
$$

and hence, if we apply the proposed compression in layer B, bounding the expected norm of the Jacobian in layer B, then full Jacobian is also bounded. Thus, constraining a network's layers should result in a more robust model. This is an interesting connection that will be evaluated in future work.

## 5 Experiments

This section presents the experiments that we have conducted and the datasets and evaluation metrics used. The implementation and training details are detailed in the Supplementary Material.

### 5.1 Datasets

To evaluate the utility of the proposed method, we conducted tests on four distinct datasets. All datasets used in this work are accessible to the public. The evaluated datasets, data augmentation, and experimental setup in this study are summarized in Table 1.

**MNIST** The Modified National Institute of Standards and Technology database (MNIST)[4] (Deng, 2012) is an extensive library of handwritten digits containing a training set of 60 000 and a test set of 10 000 grayscale $28 \times 28$ images. It is a subset of a more extensive NIST-provided collection. We used the MNIST for classification.

**CIFAR-100** The Canadian Institute for Advanced Research, 100 classes (CIFAR-100)[5] (Krizhevsky, 2009) dataset is for classification. It includes 60 000 color images with a resolution of $32 \times 32$ pixels. The images are grouped into one hundred categories, with 600 images in each category. There are 50 000 training and 10 000 test images.

**BRATS19** The Brain Tumors in Multimodal Magnetic Resonance Imaging Challenge 2019 (BRATS19)[6] (Menze et al., 2014; Bakas et al., 2017; 2018) was released in conjunction with the International Conference on Medical Image Computing and Computer Assisted Intervention (MICCAI) conference in 2019. It provides 3D multiple pre-operative magnetic resonance imaging (MRI) sequences for 285 patients. For each patient, T1-weighted (T1w), post-contrast T1-weighted (T1c), T2-weighted (T2w), and T2 Fluid Attenuated Inversion Recovery (FLAIR) scans were obtained using various procedures and scanners at a magnetic field strength of 3 T.

**SPLEEN** The Spleen Segmentation Decathlon (SPLEEN) data was part of the Medical Segmentation Decathlon[7], an open-source collection of datasets encompassing various anatomies and segmentation tasks (Simpson et al., 2019). It comprises $n = 61$ 3D computed tomography (CT) scans in which the spleen was semi-automatically annotated and was originally part of a research project on splenic volume change after chemotherapy in patients with liver metastases.

### 5.2 Experiments

We carried out the experiments on four different datasets (see Section 5.1) and three neural network architectures (see Section 5.4 in the Supplementary Material). These networks cover several network sizes and architectural families. We conducted experiments using the three different compression methods on numerous choices of layers or blocks to compress. Further, we varied the compression ratios between 1 and 1000. We investigated the impact of the proposed activation compression method with respect to the following factors.

*Impact on Memory Footprint* (Section 6.1). In this experiment, we analyzed the detailed footprint in each step in the proposed method by altering compression methods and compression ratios on two arrays with different sizes.

*Impact on Reconstructed Image* (Section 6.2). This experiment examined the impact of the compression methods at different compression ratios. Two images were selected from the datasets: one natural image from the CIFAR-100 dataset, and a 2D slice from the BRATS19 dataset. Further, we compared the mean squared error (MSE), peak signal-to-noise ratio (PSNR), and size in MB stored in GPU memory between the original and the reconstructed images. Additionally, we computed the percentage from the fraction of the compressed tensors' memory consumption over the original images' memory consumption.

*Impact on Performance* (Section 6.3). We evaluated the performance of the proposed method compared to the baseline by varying compression methods, compression ratios, and layers or blocks to compress on the four datasets. In this experiment, we compressed only batch normalization layers.

---

[4] http://yann.lecun.com/exdb/mnist/
[5] https://www.cs.toronto.edu/~kriz/cifar.html
[6] https://www.med.upenn.edu/cbica/brats-2019/
[7] http://medicaldecathlon.com/

Table 1: The datasets, augmentation, and experimental setup in this study. A depth of $D$ means that the volume shape had a varied depth.

| Material/dataset | MNIST | CIFAR-100 | BRATS19 | SPLEEN |
|---|---|---|---|---|
| Task | Classification | Classification | Segmentation | Segmentation |
| Network | LeNet | ResNet-18 | U-Net | U-Net |
| Resolution | $28 \times 28$ | $32 \times 32$ | $240 \times 240 \times 155$ | $512 \times 512 \times D$ |
| Type | grayscale | color | MRI | CT |
| #Modalities/#Channels | 1 | 3 | 4 | 1 |
| #Classes/#Masks | 10 | 100 | 3 | 1 |
| #Samples | 60 000 | 60 000 | 335 | 41 |
| Train | 42 000 | 42 000 | 235 | 29 |
| Validation | 12 000 | 12 000 | 67 | 8 |
| Test | 6 000 | 6 000 | 33 | 4 |
| Augmentation | | | | |
| Flip left-right | ✗ | ✓ | ✓ | ✗ |
| Rotation | ✗ | ✗ | ✓ | ✓ |
| Scale | ✗ | ✗ | ✓ | ✓ |
| Shift | ✗ | ✗ | ✓ | ✓ |
| Training | | | | |
| #Epochs | 50 | 70 | 70 | 70 |
| Optimizer | Adam | Adam | Adam | Adam |
| Learning rate | $1 \cdot 10^{-4}$ | $1 \cdot 10^{-4}$ | $1 \cdot 10^{-4}$ | $1 \cdot 10^{-4}$ |
| Batch size | 128 | 128 | 2 | 1 |

*Impact on Regularization* (Section 6.4). To investigate the regularization effect of the proposed method, we performed experiments using the WT method with two filters: "Haar" and "Daubechies 3" (Lee et al., 2019) on the four different datasets.

*Impact on Training Speed* (Section 6.5). We investigated the training speed (per-batch training time) when using different compression methods and compression ratios. The experiments were performed on all datasets, and only batch normalization layers were compressed.

*Impact on Allocated Memory* (Section 6.6). The allocated memory on GPU was evaluated on two architectures, ResNet18 and U-Net, using the WT (with "Haar" filters), the DCT and the ST methods. We generated real-time graphs for GPU memory usage by tracking memory allocations at each step during forward and backward propagation (see Figure 9a and Figure 9b in the Supplementary Material). To further evaluate the efficient usage of memory with respect to only the activation maps (see Equation 34) of the proposed method, we conducted a supplemental analysis with numerous settings (different networks, input shapes and batch sizes) and compared the maximum memory occupied on the GPU between methods. In addition to ResNet18 and U-Net, we included ResNet152, WideResNet101, and ViT 2D/3D. We also compared our method to those of Finder et al. (2022) and Chen et al. (2016), and a 16-bit quantization technique regarding the efficient usage of memory using another set of network architectures: EfficientNet (Tan & Le, 2019b), DeepLabV3Plus (Chen et al., 2018) with MobileNetV2 as backbone, and EDSR (Lim et al., 2017) in different settings.

### 5.3 Evaluation

We used the MSE and PSNR to evaluate the image quality after reconstruction,

$$\text{MSE}(\boldsymbol{X}, \boldsymbol{X}^l) = \frac{1}{N} \sum_{i=1}^{N} (x_i - x_i^l)^2, \tag{29}$$

$$\text{PSNR}(\boldsymbol{X}, \boldsymbol{X}^l) = 20 \cdot \log_{10}\left(\frac{\max(\boldsymbol{X})}{\sqrt{\text{MSE}(\boldsymbol{X}, \boldsymbol{X}^l)}}\right), \tag{30}$$

where $\log_{10}$ is the base-10 logarithm and $\max(\boldsymbol{X})$ is the maximum gray level in an image, $\boldsymbol{X}$. The $\boldsymbol{X}^l$ is a compressed version of $\boldsymbol{X}$. The $x_i$ is the $i$-th element of $\boldsymbol{X}$ (in any order), and $N$ is the number of pixels or voxels in the images.

We employed the soft Sørensen-Dice coefficient (DSC) loss in the segmentation tasks (Milletari et al., 2016; Vu et al., 2020a;b; 2021b;a), defined as,

$$\mathcal{L}_{\text{DSC}}(\boldsymbol{U}, \boldsymbol{V}) = \frac{-2 \sum_{i=1}^{N} u_i v_i + \epsilon}{\sum_{i=1}^{N} v_i + \sum_{i=1}^{N} v_i + \epsilon}, \tag{31}$$

where $u_i$ is the $i$-th element of a softmax output of the network, $v_i$ is the $i$-th element of a one-hot encoding of the ground truth, and $\epsilon = 1 \cdot 10^{-5}$ is a small constant added to avoid division by zero. The $\epsilon$ is present in both the numerator and the denominator to make true zero predicted masks have a score of 1.

Further, we used the categorical cross-entropy (CE) loss for the classification tasks,

$$\mathcal{L}_{\text{CE}}(\boldsymbol{U}, \boldsymbol{V}) = -\sum_{i=1}^{N} \sum_{j=1}^{K} u_i^j \cdot \log(v_i^j) \tag{32}$$

where $u_i^j$ and $v_i^j$ are the $i$-th elements for class $j$-th of network output, $\boldsymbol{U}$, and groundtruth, $\boldsymbol{V}$, respectively. The $K$ is the number of classes.

In this study, we used accuracy to evaluate the classification tasks and the negative DSC to evaluate the segmentation tasks,

$$D(\boldsymbol{U}, \boldsymbol{V}) = -\mathcal{L}_{\text{DSC}}(\boldsymbol{U}, \boldsymbol{V}). \tag{33}$$

We also introduce $\eta$, a measure of the effective usage of memory with respect to the activation maps,

$$\eta = \frac{\mathcal{M} - \mathcal{M}_{\text{BL}}}{\mathcal{M}_{\text{BL}} - \mathcal{M}_{\text{O}}} \times 100\%, \tag{34}$$

where $\mathcal{M}$, $\mathcal{M}_{\text{BL}}$, and $\mathcal{M}_{\text{O}}$ respectively represent the GPU memory that a method, the baseline, and other categories (model parameters, gradient maps, ephemeral tensor, and resident buffer; Gao et al., 2020) occupy after the forward pass. A smaller $\eta$ (in negative value) indicates a more efficient method. In particular, $\eta = 0\%$ means that no GPU memory in terms of activation maps is saved, and $\eta = -100\%$ means that the compressed activation maps consume no GPU memory.

### 5.4 Implementation Details and Training

Our method was implemented in PyTorch 1.10[8]. The experiments were run on NVIDIA Tesla K80 GPUs. Depending on the model, convergence speed, and dataset size, a single experiment was finished within a few minutes or up to several days.

We used the Adam optimizer (Kingma & Ba, 2015) in all experiments, with various initial learning rates (see Table 1 in the Supplementary Material). The number of epochs for each experiment was set to 100 for

---

[8]https://pytorch.org/

the classification task and 80 for the segmentation task. We used a small custom version of LeNet (LeCun et al., 1989) and ResNet-18 (He et al., 2016) for the MNIST and CIFAR-100 datasets, respectively. Each dataset was split into 70 % for training, 20 % for validation, and 10 % for final testing.

We used the U-Net (Ronneberger et al., 2015) for the BRATS19 and the SPLEEN datasets. Batch normalization was used after all convolutional layers in the transition block and the main network. We used the ReLU activation function. A transition block of the U-Net consists of a batch normalization layer, a ReLU layer, and a convolution layer in order to adjust (*i.e.* downsample or upsample) the channel size and the feature map size. For the ResNet18, a convolutional block included the first convolutional, the first batch normalization, a ReLU, the second convolutional, and the second batch normalization layers. In contrast, a convolutional block in the evaluated U-Net comprised a convolutional, a batch normalization, and a ReLU layer. Because there are many layers in the ResNet18 and U-Net, we conducted experiments by compressing only the batch normalization layers in a convolutional block.

Except for the SPLEEN, all datasets were standardized to zero-mean and unit variance. Before normalization, the N4ITK bias field correction (Tustison et al., 2010) was applied to the BRATS19 datasets. Samples from the SPLEEN dataset were rescaled to $[-1, 1]$.

To artificially enlarge the SPLEEN and BRATS19 dataset sizes and to increase the variability in the data, we applied on-the-fly data augmentation. Those were: horizontal flipping, rotation within a range of $-1$ to $1$ degrees, rescaling with a factor of 0.9 to 1.1, and shifting the images by $-5$ to $5$ percent in both directions.

We employed a publicly available PyTorch package, torchvision[9], for the ResNet18, ResNet152, WideResNet101, and EfficientNet models. We used a popular implementation[10] of the ViT architectures. For the other architectures, we made use of their official implementations[11][12][13].

# 6 Results and Discussion

## 6.1 Impact on Memory Footprint

Figure 4 presents the detailed footprint in each step in the proposed method (see Figure 2) using different settings in small and large arrays with input shape of $128 \times 16 \times 16 \times 1$ and $16 \times 128 \times 128 \times 128 \times 1$, respectively.

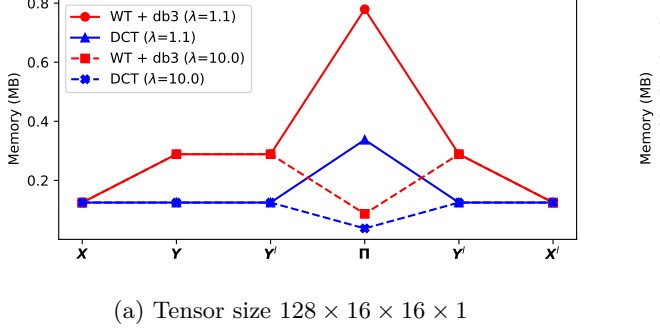
(a) Tensor size $128 \times 16 \times 16 \times 1$

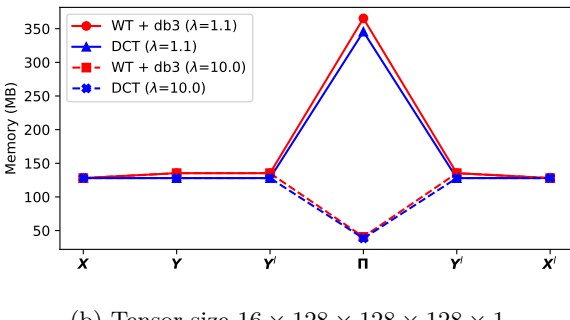
(b) Tensor size $16 \times 128 \times 128 \times 128 \times 1$

Figure 4: Illustration of footprint in each step in the proposed method (see Figure 2) over different compression ratios, $\lambda$, on two proposed methods: WT with "Daubechies 3" filter (WT + db3) and DCT.

Looking at Figure 4, it is apparent that the sparse representation in COO format (see $\mathbf{\Pi}$) incurs an overhead that becomes noticeable when the compression ratio is small, *i.e.* $\lambda = 1.1$, in both evaluated arrays. On the

---

[9]https://github.com/pytorch/vision
[10]https://github.com/lucidrains/vit-pytorch
[11]https://github.com/sanghyun-son/EDSR-PyTorch
[12]https://github.com/VainF/DeepLabV3Plus-Pytorch
[13]https://github.com/milesial/Pytorch-UNet

contrary, significant drops in allocated GPU memory can be seen from the same figure when $\lambda = 10$. More detailed analyses in terms of $\lambda$ can be found in Section 6.2 and Section 6.6.

The differences between WT and DCT in terms of memory footprint are highlighted in Figure 4. Specifically, the allocated memory on GPU of $\boldsymbol{X}$, $\boldsymbol{X}^l$, $\boldsymbol{Y}$ and $\boldsymbol{Y}^l$ of the DCT method are the same except for $\boldsymbol{\Pi}$. By way of contrast, $\boldsymbol{X}$, $\boldsymbol{Y}$, and $\boldsymbol{\Pi}$ of the WT method reserve different amounts of memory. These results are related to the shapes of the transformed matrices, $\boldsymbol{Y}$, after the WT (with padding) and DCT (without padding) are applied.

## 6.2 Impact on Reconstructed Image

Figure 5 (top) shows the original image and its compressed, thresholded, and decompressed version at different compression ratios using four methods: WT with "Haar" filter (WT + "haar"), WT with "Daubechies 3" filter (WT + "db3"), DCT, and ST. Further, Figure 5 (bottom) compares the MSE, PSNR, size in megabytes (MB) in memory, and the percent of memory used by the compressed image over the original image.

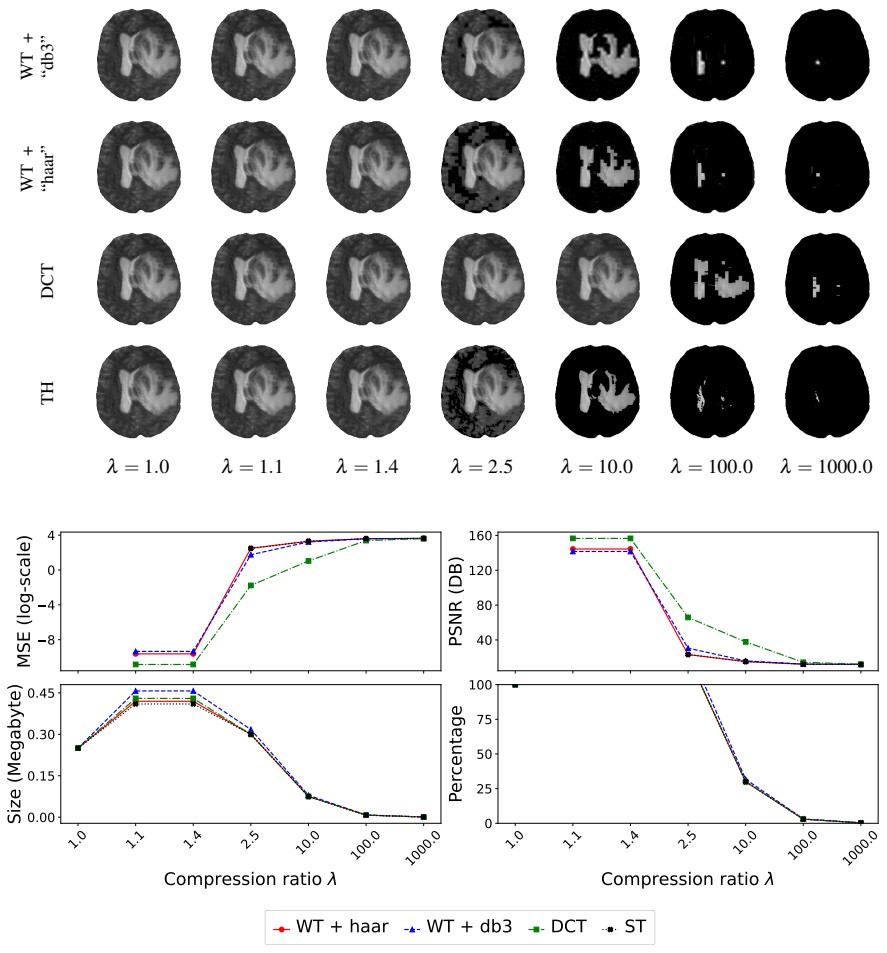

Figure 5: Quality of reconstructed images over different compression ratios, $\lambda$, on four proposed methods: WT with "Haar" filter (WT + "haar"), WT with "Daubechies 3" filter (WT + "db3"), DCT, and ST. The $\lambda = 1$ represents the original image (or baseline, *i.e.* without compression). **Top**: Illustration of original and uncompressed (or reconstructed) images. **Bottom**: Comparison of the MSE, PSNR, size in MB stored in the memory, and the percentage computed as a fraction of the memory consumption of the compressed image over the original image.

From Figure 5, we see that when the compression ratio is smaller than 100, the DCT method outperforms the others in terms of the MSE and PSNR. Looking at Figure 5 (bottom), it is apparent that the size in MB in memory is comparable for all methods at all compression ratios. This result can be explained by the fact that the thresholding technique used in this work is top-$k$, meaning that a certain amount of non-zero elements are expected to be kept when the same compression ratio is used in all methods.

Another observation emerges from Figure 5 (top) is that the reconstructed images using the WT compression method become mostly black, but not blurry as seen in other works (*e.g.*, Haar, 1909; Zweig, 1976; Mallat, 1989) when the compression ratio increases. There are two possible explanations for that behavior: (1) the thresholding technique used in this work is top-$k$ and (2) we threshold all sub-bands, including the low-frequency component $\boldsymbol{Y}^{(2)}_{\text{low,low}}$ (see Figure 1).

It can also be seen from Figure 5 (bottom) that when the compression ratio is smaller than 10, the reconstructed image sizes of all methods are larger than the size of the original image. The behavior related to the size can be explained by the fact that the sparse version of the thresholded tensor, in the COO format, is ineffective in terms of memory allocation. A simple solution for the proposed method would be to compare the sparse tensor's size to the original tensor's size and simply store and use the smallest one. This would eliminate the dense and inverse transform steps when there is no benefit of converting to the sparse format and use the reconstructed tensor directly when computing the gradients. This solution saves some computations but does not save any memory.

## 6.3   Impact on Performance

Figure 6 illustrates the accuracies and their standard errors on the MNIST and CIFAR-100 data and the DSC scores and their standard errors on the SPLEEN and BRATS19 data, for different compression ratios. The title of each sub-figure for the MNIST dataset (Figure 6a) indicates which layer was compressed. For example, "conv1" means that only the first convolutional layer was compressed, while "conv1-relu1" means that the first convolutional and the first ReLU layers were compressed. For the other three datasets, we used "block0" to indicate that all batch normalization layers in the first convolutional layer were compressed. Similarly, we used "block0-block1" to specify that all batch normalization layers in the first and the second convolutional layers were compressed, and so on.

It is apparent from Figure 6a that the performance of the DCT method on the MNIST dataset is comparable to the baseline's when more than one layer was compressed. Another striking observation to emerge from Figure 6a is that the accuracies on the MNIST dataset go down gradually starting from the compression ratio of 10.

It can be seen in Figure 6b that the accuracies of the DCT and WT methods are on a level with the baseline's when the batch normalization layers are compressed in block zero, one, and two. However, the curves seem to decline slowly from block three to block four and drop dramatically when more than three blocks are compressed, specifically from about $\lambda = 2.5$. There are several possible explanations for these results. First, the resolutions of features maps in the third block and last block of ResNet18 are $256 \times 8 \times 8$ and $512 \times 4 \times 4$, respectively. Therefore, when $\lambda$ is larger than 2.5, very few details are left causing the network to fail to learn (see "Block3" and "Block4" in Figure 6b). Second, when all blocks are compressed, the information, that is carried from prior blocks ("Block0" to "Block2") might be insufficient to make the last two compressed blocks ("Block3" and "Block4") extract relevant features. In other words, this may be explained by collective information loss.

What is interesting in Figure 6c and Figure 6d is that the DSC scores of the DCT methods are on par with the baseline's when the compression ratio is less than or equal to 100. It seems that $\lambda = 100$ is a delicate threshold to obtain satisfactory performance for the DCT method.

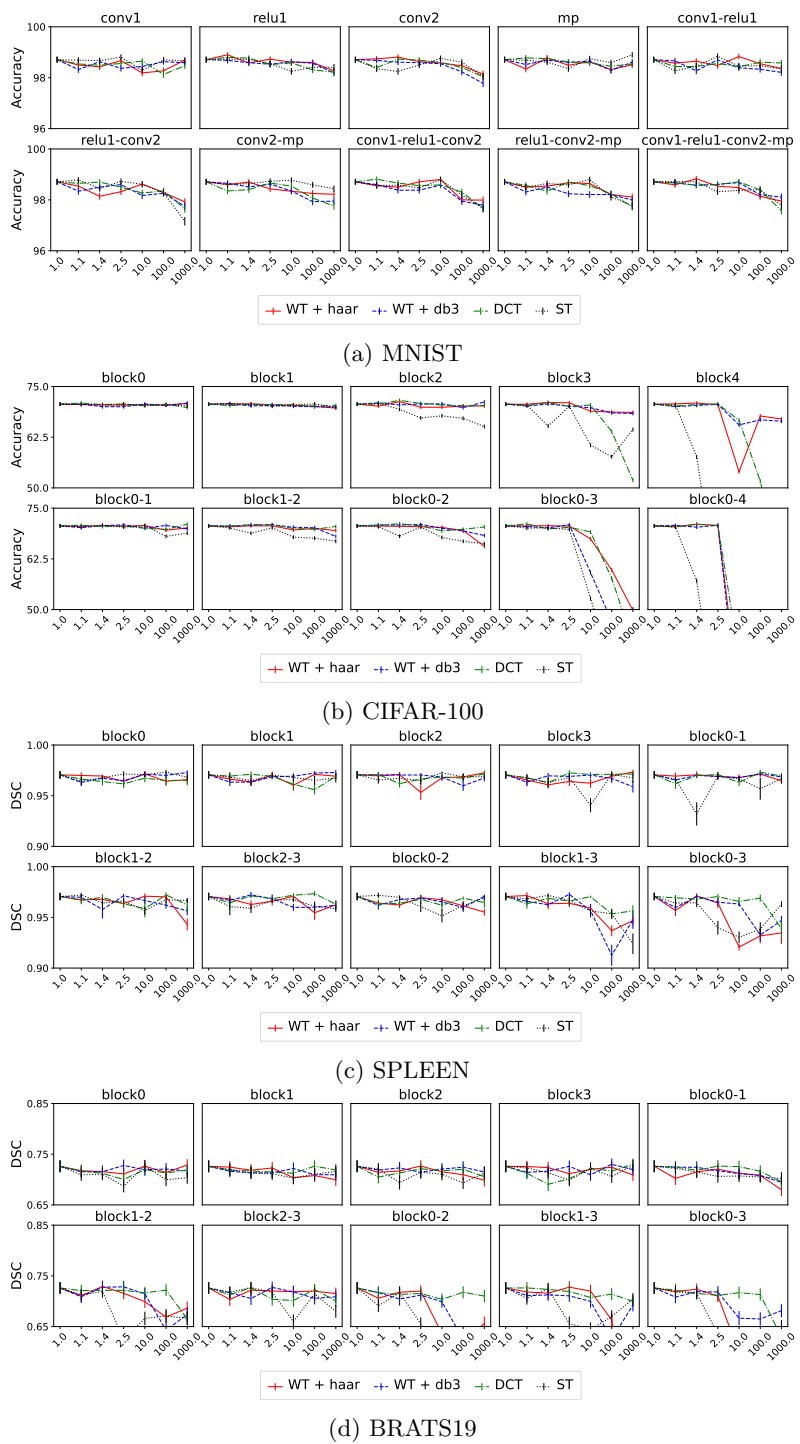

Figure 6: Performance over different compression ratios on four different datasets using four proposed methods: WT with "Haar" filter (WT + "haar"), WT with "Daubechies 3" filter (WT + "db3"), DCT, and ST. The $x$-axis shows the compression ratio, $\lambda$. The "mp" stands for max pooling layer in the MNIST. In the MNIST, "conv1", "relu1", and so on, show which layer is compressed. For the others, "block0", "block1", and so on indicates which convolutional block the batch normalization layers are compressed.

### 6.4 Impact as Regularization

Figure 7 illustrates the regularization effect of the proposed method and shows the accuracies and DSC scores and their standard errors. The title of each sub-figure in Figure 7 denotes the compressed layer. The "conv1" is the first convolutional layer of the network, the "conv2" is the second convolutional layer, and so on.

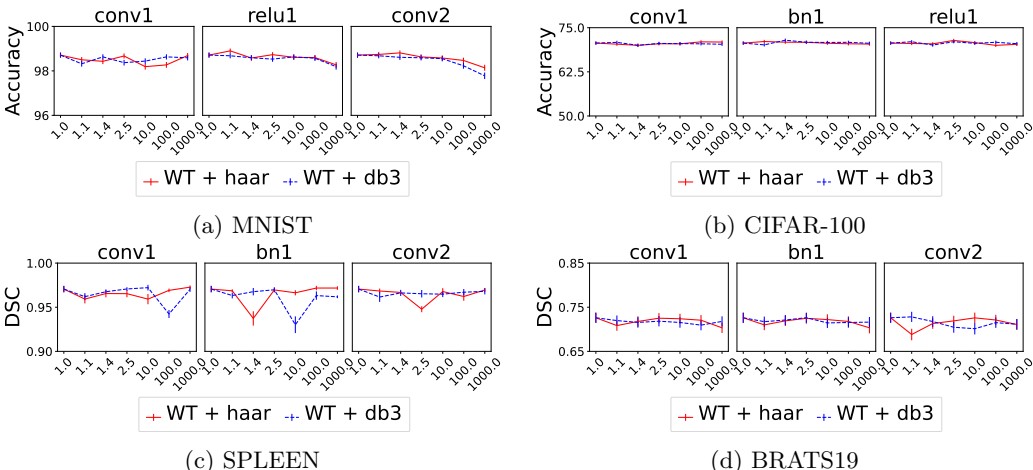

(a) MNIST            (b) CIFAR-100

(c) SPLEEN           (d) BRATS19

Figure 7: Illustration of regularizing effect. Performance over different compression ratios on four different datasets using two proposed methods: WT with "Haar" filter (WT + "haar") and WT with "Daubechies 3" filter (WT + "db3"). The $x$-axis shows the compression ratio, $\lambda$.

From Figure 7, we see that: (1) the accuracy on the CIFAR-100 and the DSC on the BRATS19 remain unchanged when the compression ratios increase on all evaluated compressed layers, and (2) the accuracy on the MNIST and the DSC on the SPLEEN fluctuate when the compression ratios increase.

### 6.5 Impact on Training Speed

Figure 8 illustrates the training time per batch (in seconds) on the four datasets for different compression ratios in the ResNet18 and U-Net. In all cases, we can again see that a compression ratio of 10 might be a good value for the WT methods, because the batch durations did not increase much compared to the baseline.

Figure 8 illustrates a slight decrease in the batch duration in all cases when the compression ratio increases. Additionally, it is clear that the DCT method is computationally more expensive than the other methods. This is because the DCT method splits an image into a large number of $16 \times 16$ blocks, and processes each block separately, making the computation inefficient. An alternative could be to perform the DCT on the whole image instead of splitting the image into blocks and applying the DCT on them.

As can be seen in Figure 8a, there is a considerable variance in batch duration when different layers are compressed on the MNIST dataset when the WT and ST methods are used. Specifically, the batch durations when we compress the first convolutional ("conv1") or max pooling layers ("mp") are on the same level as the baseline. The batch durations jump dramatically for the other cases. For the CIFAR-100 dataset that can be seen in Figure 8b, the batch duration increases slightly for the WT and ST methods when block zero ("block0") is compressed.

As shown in Figure 8c and Figure 8d, there is only a small difference in the batch durations when comparing the baseline to the WT and ST methods at different compression ratios. We can also see from  Figure 8c and Figure 8d that the batch duration when batch normalization layers are compressed in the first convolutional block ("block0") is 14% longer than that in the third convolutional block ("block2").

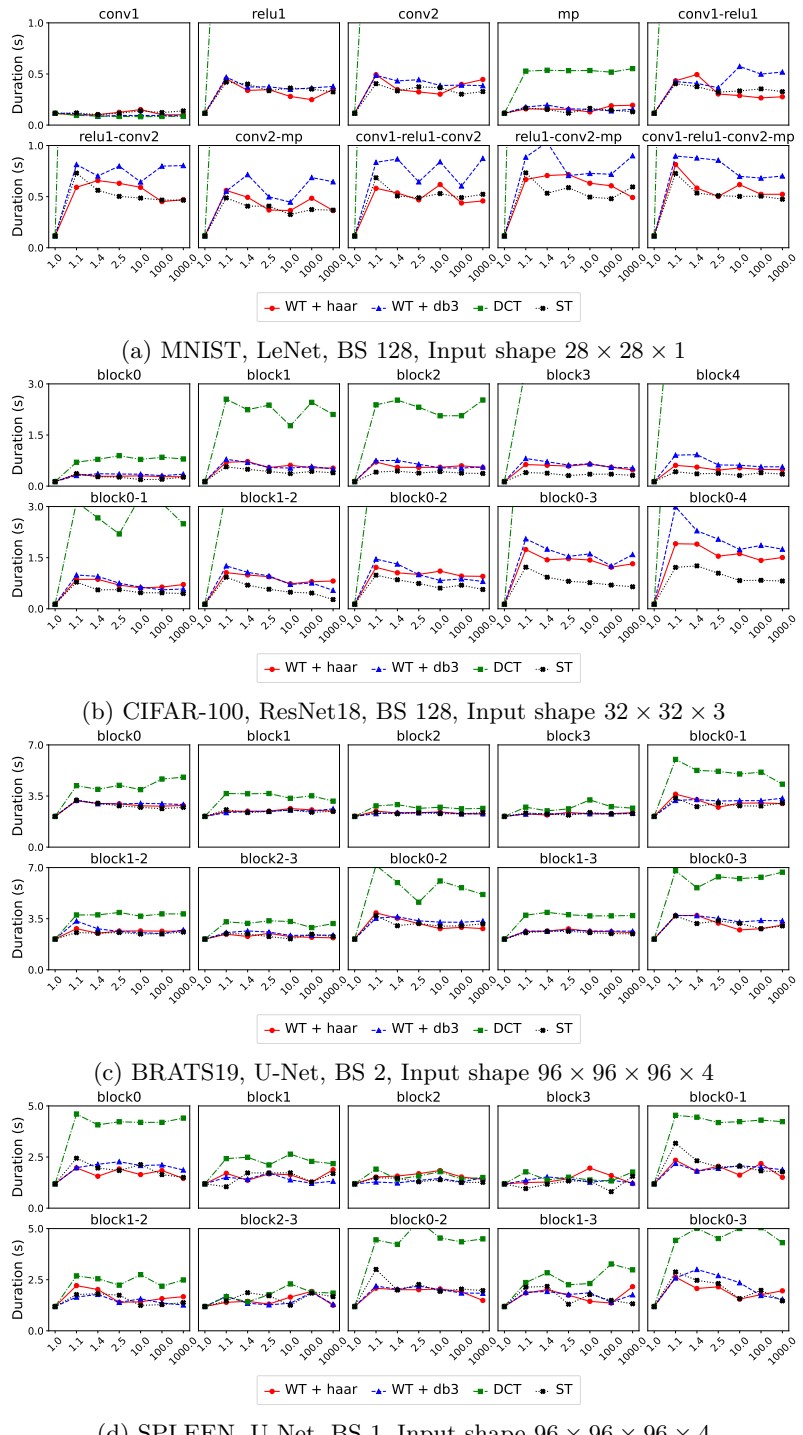

Figure 8: Batch duration over different compression ratios on four different datasets using four proposed methods: WT with "Haar" filter (WT + "haar"), WT with "Daubechies 3" filter (WT + "db3"), DCT, and ST. The $x$-axis shows the compression ratio, $\lambda$. "BS" stands for batch size. The "mp" stands for max pooling layer in the MNIST. In the MNIST, "conv1", "relu1", and so on, show which layer is compressed. For the others, "block0", "block1", and so on indicates which convolutional block the batch normalization layers are compressed.

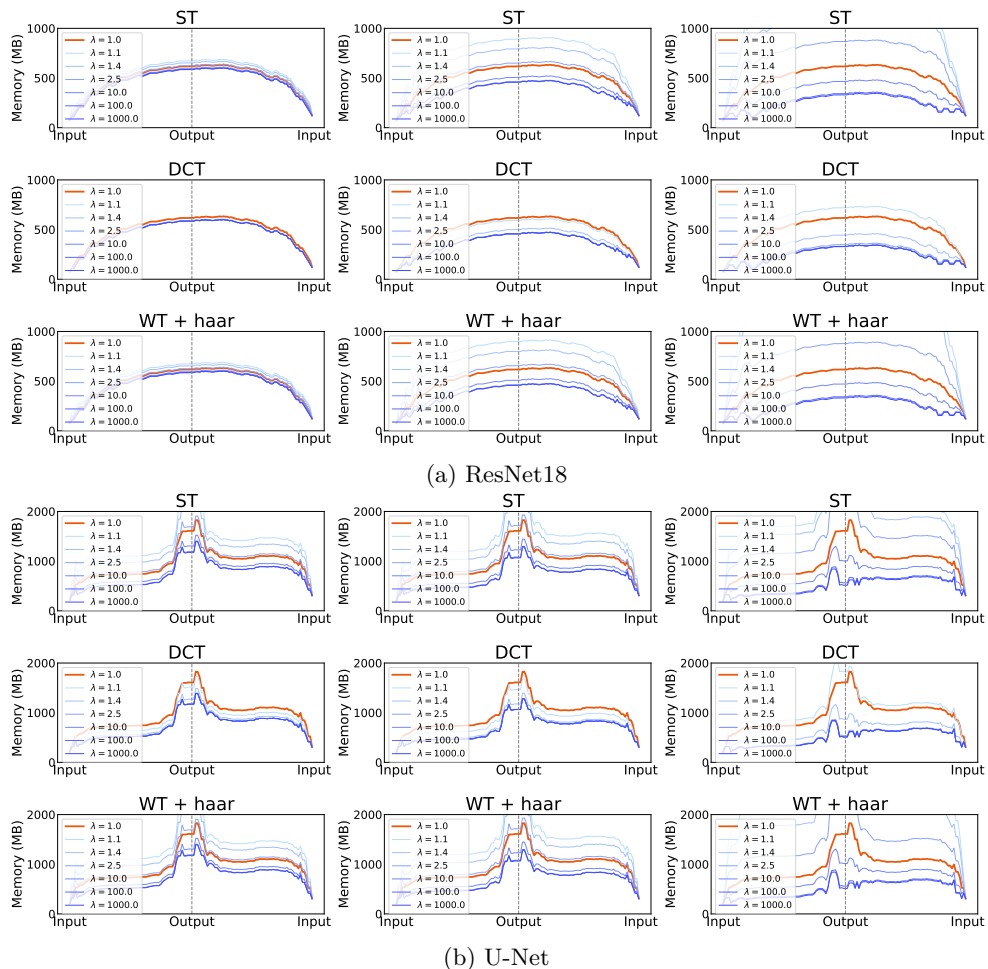

Figure 9: Illustration of the memory allocation at each step ($x$-axis) during the forward and backward pass on **(a)** ResNet18 and **(b)** U-Net at the compression ratio of 10. The $x$-axis shows the compression ratio, $\lambda$. The input shape and batch size of ResNet18 are $32 \times 32 \times 3$ and 128, respectively. The input shape and batch size of U-Net are $96 \times 96 \times 96 \times 3$ and 1, respectively. The vertical dashed line denotes where the forward-propagation ends and the back-propagation begins. **Left**: batch normalization layers in the first convolutional block are compressed. **Middle**: batch normalization layers in the first two convolutional blocks are compressed. **Right**: convolutional layers, batch normalization layers, and ReLU layers in the first two convolutional blocks are compressed.

## 6.6 Impact on Allocated Memory

Figure 9a and Figure 9b illustrate the memory allocation in MB at each step during the forward followed by the backward pass on ResNet18 and U-Net, respectively, with $\lambda = 10$ using the DCT, the WT with "Haar" filter and the ST methods. In the first two columns in both figures, the "first convolutional block (BN)" and "first two convolutional blocks (BN)" show at which block the batch normalization layers are compressed. For the last column, "first two convolutional blocks (Conv. + BN + ReLU)" indicates that three layers, including convolutional, batch normalization, and ReLU, are compressed in the first two convolutional blocks. The vertical dashed line shows where the forward-propagation ends and where the back-propagation begins.

As shown in both Figure 9a and Figure 9b, the GPU memory usage is higher than the baseline when the compression ratio is smaller than 10 for the WT and ST methods and for both the ResNet18 and U-Net. An explanation for this is that very little information is removed when the compression ratio is small, for

Table 2: Illustration of memory allocated (in MB) on the GPU when DCT is used with compression ratio 100. "Comp. layers" denote which layers or modules were compressed. The "block0" denotes that activation maps were compressed at the first convolutional block, the "block0–1" denotes activation maps compressed at the first two convolutional blocks, and so on. The "BN", "Conv." and "ReLU" denote batch normalization, convolutional, and ReLU layers, respectively. The "Attention" and "Feedforward" are modules introduced in ViT. "Others" indicates the GPU memory occupied by all categories except for the activation maps. The "BL" shows the GPU memory occupied by all categories (including the activation maps) on the baseline (without layers compressed). The % values denote the memory efficiency, $\eta$, in negative percentage with respect to only the activation maps (a lower negative value is better). ResNet152* and ResNet152 denote two versions of ResNet152 that use BasicBlock (as in ResNet18) and BottleneckBlock, respectively.

| Network | Comp. layers | Others $(\mathcal{M}_{\mathrm{O}})$ | Baseline $(\mathcal{M}_{\mathrm{BL}})$ | DCT + $\lambda$ = 100 $(\mathcal{M})$ | | | | |
|---|---|---|---|---|---|---|---|---|
| | | | | block0 | block0–1 | block0–2 | block0–3 | block0–4 |
| ResNet18 Batch size 128 Res. $32 \times 32 \times 3$ | BN Conv. + BN Conv. + BN + ReLU | 133 | 617 | 586 (-6%) 586 (-6%) 595 (-5%) | 462 (-32%) 497 (-25%) 425 (-40%) | 400 (-45%) 454 (-34%) 360 (-53%) | 369 (-51%) 431 (-38%) 327 (-60%) | 353 (-55%) 418 (-41%) 311 (-63%) |
| ResNet18 Batch size 512 Res. $32 \times 32 \times 3$ | BN Conv. + BN Conv. + BN + ReLU | 402 | 2338 | 2213 (-6%) 2213 (-6%) 2252 (-4%) | 1726 (-32%) 1867 (-24%) 1575 (-39%) | 1482 (-44%) 1698 (-33%) 1321 (-53%) | 1360 (-51%) 1610 (-38%) 1196 (-59%) | 1298 (-54%) 1563 (-40%) 1133 (-62%) |
| ResNet18 Batch size 128 Res. $64 \times 64 \times 3$ | BN Conv. + BN Conv. + BN + ReLU | 402 | 2338 | 2214 (-6%) 2214 (-6%) 2253 (-4%) | 1727 (-32%) 1869 (-24%) 1576 (-39%) | 1483 (-44%) 1701 (-33%) 1323 (-52%) | 1361 (-50%) 1615 (-37%) 1199 (-59%) | 1299 (-54%) 1565 (-40%) 1136 (-62%) |
| U-Net Batch size 1 Res. $96 \times 96 \times 96 \times 4$ | BN Conv. + BN Conv. + BN + ReLU | 458 | 1630 | 1206 (-36%) 1004 (-53%) 803 (-71%) | 1102 (-45%) 836 (-68%) 583 (-89%) | 1076 (-47%) 793 (-71%) 527 (-94%) | 1069 (-48%) 782 (-72%) 514 (-95%) | *n.a* *n.a* *n.a* |
| U-Net Batch size 2 Res. $96 \times 96 \times 96 \times 4$ | BN Conv. + BN Conv. + BN + ReLU | 827 | 3158 | 2321 (-36%) 1916 (-53%) 1509 (-71%) | 2112 (-45%) 1579 (-68%) 1072 (-89%) | 2060 (-47%) 1494 (-71%) 962 (-94%) | 2046 (-48%) 1474 (-72%) 934 (-95%) | *n.a* *n.a* *n.a* |
| U-Net Batch size 1 Res. $128 \times 128 \times 128 \times 4$ | BN Conv. + BN Conv. + BN + ReLU | 962 | 3724 | 2732 (-36%) 2250 (-53%) 1771 (-71%) | 2482 (-45%) 1853 (-68%) 1251 (-90%) | 2421 (-47%) 1753 (-71%) 1120 (-94%) | 2407 (-48%) 1728 (-72%) 1088 (-95%) | *n.a* *n.a* *n.a* |
| ResNet152* Batch size 128 Res. $32 \times 32 \times 3$ | BN Conv. + BN Conv. + BN + ReLU | 312 | 2413 | 2382 (-1%) 2382 (-1%) 2383 (-1%) | 2196 (-10%) 2201 (-10%) 2016 (-19%) | 1947 (-22%) 1961 (-22%) 1544 (-41%) | 1389 (-49%) 1420 (-47%) 452 (-93%) | 1365 (-50%) 1398 (-48%) 410 (-95%) |
| ResNet152 Batch size 128 Res. $32 \times 32 \times 3$ | BN Conv. + BN Conv. + BN + ReLU | 738 | 6287 | 6956 (0%) 6956 (0%) 6957 (0%) | 6399 (-9%) 6414 (-9%) 5985 (-16%) | 5609 (-22%) 5650 (-21%) 4492 (-40%) | 3909 (-49%) 4005 (-48%) 1179 (-93%) | 3827 (-51%) 3926 (-49%) 1036 (-95%) |
| WideResNet101 Batch size 128 Res. $32 \times 32 \times 3$ | BN Conv. + BN Conv. + BN + ReLU | 1000 | 6824 | 4880 (-33%) 4880 (-33%) 4880 (-33%) | 4522 (-40%) 4542 (-39%) 3526 (-57%) | 4294 (-43%) 4334 (-43%) 3192 (-62%) | 3811 (-52%) 3899 (-50%) 1314 (-95%) | 3869 (-51%) 3960 (-49%) 1275 (-95%) |
| ViT Batch size 256 Res. $256 \times 256 \times 3$ | Attention + Feedforward | 467 | 6299 | 5331 (-17%) | 4378 (-33%) | 3427 (-49%) | 2479 (-66%) | 1532 (-82%) |
| ViT 3D Batch size 32 Res. $16 \times 128 \times 128 \times 3$ | Attention + Feedforward | 631 | 6333 | 5975 (-6%) | 5604 (-13%) | 4235 (-37%) | 2867 (-61%) | 1498 (-85%) |

instance, when $\lambda = 1.1$. Thus, the overhead of the sparse representation (in COO format) begins to show in this case, as the tensors are not actually particularly sparse.

From Figure 9a, we see that when batch normalization layers are compressed in only the first block ("block0") (left column of Figure 9a), the GPU usage for the activation maps decreases by only 5%. There is a substantial drop (about 25%) in GPU memory usage at a compression ratio of 10 when batch normalization layers are compressed in the first two convolutional blocks (middle column of Figure 9a). The GPU memory usage plummets by 35% at a compression ratio of 10 when three layers, including convolutional, batch normalization, and ReLU, are compressed in the first two convolutional blocks (right column of Figure 9a).

It can be seen from Figure 9b that about 30% of the GPU memory usage is saved when batch normalization layers are compressed in the first convolutional block (left column) and the first two convolutional blocks (right column) in U-Net with the compression ratio of 10. Again, the GPU memory use falls significantly (about 80%) at a compression ratio of 10 when three layers, including convolutional, batch normalization, and ReLU, are compressed in the first two convolutional blocks (right column).

Table 2 presents the memory allocated (in MB) on the GPU using DCT at compression ratio of 100 when different layers ("Comp. layers") are compressed at different convolutional blocks. The "Block0" represents evaluated layers' activation maps that are compressed at the first convolutional block, the "Block0-1" represents evaluated layers' activation maps that are compressed at the first two convolutional blocks, and so on. The "BN", "Conv.", and "ReLU" denote batch normalization, convolutional, and ReLU layers, respectively. The "Other" indicates the GPU memory occupied by all categories except for the activation maps. The "BL" shows the GPU memory occupied by all categories (including the activation maps) on the baseline (without layers compressed). The percent values denote the memory efficiency, $\eta$, in percentage with respect to only the activation maps (see Equation 34).

It can be seen from Table 2 that there are remarkable rises in GPU memory efficiency when layers' activation maps (for all cases) are compressed in the first convolutional ("Block0") compared to in the first two convolutional blocks ("Block0–1"), from 5% to around 40% in ResNet18 and from 71% to 89% in U-Net. The jumps are insignificant when layers' activation maps are compressed from the third convolutional block forward ("Block0–2", "Block0–3", and "Block0–4"). Compared to ResNet18 and U-Net, Table 2 reveals that there have been significant rises in GPU memory efficiency when layers' activation maps are compressed from the third convolutional block forward ("Block0–3") in ResNet152* (from 41% to 93%), ResNet152 (from 40% to 93%) and WideResNet101 (from 62% to 95%). In contrast, the figures are proportional to the number of compressed blocks in ViT architectures.

From Table 2, we can see that, with respect to the GPU memory occupied by only activation maps, the proposed method using DCT saves about 40% when convolutional, batch normalization, and ReLU layers are compressed in the first two convolutional blocks in the ResNet18. The figures then rise to 62–63% when these layers are compressed in all convolutional blocks in all evaluated settings of ResNet18. The figures for the U-Net are even higher, going from 36 to 48%, from 53 to 72%, and from 71 to 95% when "BN", "Conv. + BN" and "Conv. + BN + ReLU" layers are compressed, respectively. ResNet152*, ResNet152, and WideResNet101 exhibit the same patterns as U-Net. However, when the "Block0" is compressed, only 0-1% of the GPU memory occupied by activation maps is saved by two ResNet152 variants. This result may be attributed to the significantly smaller number of layers in the first block of these architectures compared to the whole network.

As shown in Table 2, the proposed compression method using DCT is more memory-efficient in the higher dimensional activation maps when comparing $\eta$ in evaluated architectures. For instance, the proposed method saves GPU memory usage by about 5–6% in ResNet18, 36–71% in U-Net, around 1% in two ResNet152 variants, and 33% in WideResNet101 when evaluating layers of the first convolutional block. The figures increase from 38 to 60% in ResNet18, from 48 to 95% in U-Net, and from about 52% to 94% in larger models (ResNet152 and WideResNet101) when evaluating layers of the first four convolutional blocks.

What is surprising from Table 2 is that compressing more layers results in more GPU memory saving for the case of U-Net, but not for the ResNet family (ResNet18, ResNet152*, ResNet152 and WideResNet101). In particular, compressing convolutional and batch normalization layers ("Conv. + BN") consumes more memory than compressing only batch normalization layers ("BN"), but compressing three layers ("Conv. + BN + ReLU") reduces the memory consumption. The reason for this unexpected behavior is likely due to the presence of addition operation in ResNet block that prevents the original tensor from being released from the GPU *i.e.* PyTorch stores both original and compressed tensors on the GPU.

Table 3 compares the memory allocated on the GPU using different methods with different parameters. Instead of employing 32-bit in feature maps and network parameters as in other methods, we analyze the quantization technique with 16-bit. In the "Checkpoint" method (Chen et al., 2016), $s$ stands for the number of segments. In WCC, 4-bit weights and 8-bit activations are selected, and $\omega$ denotes the compression rate. In this table, we experiment with our method with different compression ratios, $\lambda$.

Table 3: Illustration of memory allocated (in MB) on the GPU when different methods are used. The $\mathcal{M}_\mathrm{O}$ indicates the GPU memory occupied by all categories except for the activation maps. The $\mathcal{M}_\mathrm{BL}$ shows the GPU memory occupied by all categories (including the activation maps) on the baseline (without layers compressed). The $\mathcal{M}$ shows the GPU memory that a method occupy. The % values denote the memory efficiency, $\eta$, in negative percentage with respect to only the activation maps (a lower negative value is better). In "Checkpoint" method, $s$ stands for the number of segments. The $\omega$ denotes the compression rate, while 4-bit weights and 8-bit activations are selected in WCC.

| Configuration | | EfficientNet B0 Batch size 4 Res. $640 \times 640 \times 3$ | | | EfficientNet B4 Batch size 2 Res. $640 \times 640 \times 3$ | | | EfficientNet B8 Batch size 1 Res. $640 \times 640 \times 3$ | | |
|---|---|---|---|---|---|---|---|---|---|---|
| | | $\mathcal{M}_\mathrm{O}$ | $\mathcal{M}_\mathrm{BL}$ | $\mathcal{M}(\eta)$ | $\mathcal{M}_\mathrm{O}$ | $\mathcal{M}_\mathrm{BL}$ | $\mathcal{M}(\eta)$ | $\mathcal{M}_\mathrm{O}$ | $\mathcal{M}_\mathrm{BL}$ | $\mathcal{M}(\eta)$ |
| Quantization | 16-bit | 20 | 1017 | n.a | 25 | 939 | n.a | 40 | 888 | n.a |
| Checkpoint (Chen et al., 2016) | $s = 1$ | | | 1196 (-42%) | | | 1130 (-41%) | | | 1098 (-40%) |
| | $s = 2$ | | | 310 (-86%) | | | 322 (-85%) | | | 345 (-84%) |
| | $s = 3$ | 39 | 2023 | 168 (-93%) | 49 | 1871 | 185 (-93%) | 79 | 1781 | 216 (-92%) |
| | $s = 4$ | | | 140 (-95%) | | | 151 (-94%) | | | 177 (-94%) |
| | $s = 5$ | | | 117 (-96%) | | | 130 (-96%) | | | 161 (-95%) |
| WCC (Finder et al., 2022) | None | | | 7120 (257%) | | | 6580 (258%) | | | 6179 (258%) |
| | $\omega = 50\%$ | | | 5629 (182%) | | | 5121 (178%) | | | 4843 (180%) |
| | $\omega = 25\%$ | 39 | 2023 | 4876 (144%) | 49 | 1871 | 4418 (140%) | 79 | 1781 | 4193 (142%) |
| | $\omega = 12.5\%$ | | | 4493 (124%) | | | 4051 (120%) | | | 3834 (121%) |
| | $\omega = 6.25\%$ | | | 4306 (115%) | | | 3862 (109%) | | | 3645 (110%) |
| Ours (DCT) | $\lambda = 1.1$ | | | 7427 (272%) | | | 6805 (271%) | | | 6387 (271%) |
| | $\lambda = 1.4$ | | | 5786 (190%) | | | 5314 (189%) | | | 5013 (190%) |
| | $\lambda = 2.5$ | 39 | 2023 | 3327 (66%) | 49 | 1871 | 3075 (66%) | 79 | 1781 | 2973 (70%) |
| | $\lambda = 10$ | | | 872 (-58%) | | | 806 (-58%) | | | 786 (-58%) |
| | $\lambda = 100$ | | | 121 (-96%) | | | 124 (-96%) | | | 149 (-96%) |
| | $\lambda = 1000$ | | | 48 (-100%) | | | 57 (-100%) | | | 87 (-100%) |

| Configuration | | DeepLabV3Plus + MobileNetV2 Batch size 32 Res. $128 \times 128 \times 3$ | | | EDSR Batch size 4 Res. $48 \times 48 \times 3$ | | |
|---|---|---|---|---|---|---|---|
| | | $\mathcal{M}_\mathrm{O}$ | $\mathcal{M}_\mathrm{BL}$ | $\mathcal{M}(\eta)$ | $\mathcal{M}_\mathrm{O}$ | $\mathcal{M}_\mathrm{BL}$ | $\mathcal{M}(\eta)$ |
| Quantization | 16-bit | 160 | 1373 | n.a | 92 | 487 | n.a |
| Checkpoint (Chen et al., 2016) | $s = 1$ | | | 2683 (-2%) | | | 941 (0%) |
| | $s = 2$ | | | n.a | | | 644 (-38%) |
| | $s = 3$ | 318 | 2722 | n.a | 166 | 943 | 536 (-52%) |
| | $s = 4$ | | | n.a | | | 491 (-58%) |
| | $s = 5$ | | | n.a | | | 455 (-63%) |
| WCC (Finder et al., 2022) | None | | | 8295 (232%) | | | 3288 (302%) |
| | $\omega = 50\%$ | | | 6512 (158%) | | | 3280 (301%) |
| | $\omega = 25\%$ | 318 | 2722 | 5612 (120%) | 166 | 943 | 3277 (300%) |
| | $\omega = 12.5\%$ | | | 5165 (102%) | | | 3275 (300%) |
| | $\omega = 6.25\%$ | | | 4929 (92%) | | | 3274 (300%) |
| Ours (DCT) | $\lambda = 1.1$ | | | 6296 (149%) | | | 2781 (237%) |
| | $\lambda = 1.4$ | | | 5307 (108%) | | | 2400 (188%) |
| | $\lambda = 2.5$ | 318 | 2722 | 3209 (20%) | 166 | 943 | 1443 (64%) |
| | $\lambda = 10$ | | | 1053 (-69%) | | | 488 (-59%) |
| | $\lambda = 100$ | | | 390 (-97%) | | | 199 (-96%) |
| | $\lambda = 1000$ | | | 325 (-100%) | | | 170 (-99%) |

From Table 3, we see the following. First, the 16-bit quantization technique reduces the allocated memory by half of that as in the default configuration *i.e.* 32-bit. For example, $\mathcal{M}_{BL}$ is 2023 and 1017 in the EfficientNet B0 when 32-bit and 16-bit are used, respectively. Second, the "Checkpoint" method decreases the GPU usage by large margins in all models except for the combination of DeepLabV3Plus and MobileNetV2 where the sole choice for the number of segments, $s$, is 1. A possible explanation for this might partly be explained by the fact that the official PyTorch implementation for this method considers MobileNetV2, which is based on inverted residual blocks, as a single module. Third, the WCC consumes more GPU memory than the baseline in all cases. These results can be explained by the fact that WCC trades GPU memory for computational cost (Finder et al., 2022). Last, in our method, the figures are positive when $\lambda$ are smaller than 2.5, but become negative afterward. A possible explanation was mentioned earlier that very little is saved when the compression ratio is small, because of the overhead in the sparse representation.

The "Checkpoint" method optimizes GPU memory usage and works well on most evaluated architectures. However, it has a major weakness with architectures based on skip connections (*e.g.*, U-Net) or residual blocks (*e.g.*, ResNet family). To be precise, the "Checkpoint" method fails easily if the feature maps that are checkpointed lie in the middle of residual blocks. In contrast, the proposed method tackles this challenge comfortably and naturally.

## 7    Conclusion

We have presented a novel method for intermediate feature map compression in DNNs that reduces the memory footprint on a GPU. The study was performed on classification and semantic segmentation tasks. We evaluated compression based on the WT, the DCT, and a ST method. We demonstrated that the proposed method reduces the activation map memory usage by large margins (by up to 95%) while maintaining the performance compared to the baseline, and especially so when using DCT compression. In addition, we showed in theory that the proposed activation compression method induces a gradient penalty—a constraint on the expected squared Frobenius norm of the compressed layer's weight gradients. More experiments are needed for in-depth characterization of the regularization effect.

The evidence reviewed here suggests that top-$k$ is not an adequate approach to threshold sub-bands decomposed using the WT since its performances are less stable and lower compared to when using the DCT. Hence, further work could determine more effective thresholding techniques for compression with the WT. Future work could also include determining the thresholds or the transforms adaptively, *i.e.* automatically finding optimal ones for a particular dataset.

Though the proposed compression method is able to save much GPU memory in terms of activation maps, the sparse tensor technique appears to have a large overhead. Future research could include other compression methods, such as Fourier transform and Huffman coding, with the purpose of not only saving even more GPU memory but also speeding up the training. Another research question that could be asked includes comparing different sparse representations.

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
