# OpenReview forum: "Compressing the Activation Maps in Deep Convolutional Neural Networks and Its Regularizing Effect"
_TMLR — Accepted by TMLR_

### Review · Reviewer_X2os · 2023-10-16

**Summary Of Contributions:**

This paper proposes a novel approach for compressing high-dimensional activation maps obtained using deep learning architectures. The proposed approach was compared with Wavelet Transform, Discrete Cosine Transform, and Simple Thresholding in two classification tasks for natural images and two semantic segmentation tasks for medical images. In the experimental analyses, the proposed method enables reduction of the memory usage for activation maps.  In the mathematical analyses, regularization property of the proposed approach was explored.

**Audience:**

Yes

**Broader Impact Concerns:**

No concerns on the ethical implications.

**Claims And Evidence:**

No

**Requested Changes:**

Some of the major/minor comments:

- Section 3 needs to be revised. More precisely, the novelty of the proposed approach is not well defined. The section starts by reviewing different transformation methods. Then, thresholding and sparsification/reconstruction methods are employed. Since this pipeline (decomposition, pruning and sparsification) is popularly utilized for data compression, the novelty of the proposed approach should be explained more clearly.

- Could you please explain the sparse encoding eqn. (12) and dense decoding eqn. (13) methods?

- In the theoretical analyses (e.g. Theorems), sparsification/reconstruction was not considered. How do they affect the theoretical results?

- How do you employ W of WT in (8) and B of DCT in (9)? Do you further optimize them or use fixed random matrices?

- Sparse computations on GPU can be computationally complex indeed since they are not supported by many libraries and their implementation can be challenging. Could you please provide implementation details?

- Could you please provide a detailed footprint analysis of each step in the pipeline?

- How does reconstruction increase the footprint?

- In the motivation, larger DNNs such as transformers have been considered. Could you please extend the analyses with larger ResNets (e.g with layers >50), WideResnets and Transformers?

- Ablations implementing different pruning and sparsification methods should be provided as well.

**Strengths And Weaknesses:**

The proposed pipeline has been well employed in practice. However, the proposed analyses are interesting.

Briefly, there are a few major weaknesses:

1. The novelty of the proposed pipeline is not clear. Several implementation and algorithmic details are also missing.

2. The theoretical analyses should be improved considering all of the proposed methods, e.g. including sparsification.

3. Experimental analyses should be improved using additional models and with ablations.

---

> ### Author Response · Authors · 2023-12-06
> **Reply to Reviewer X2os [1]**
>
> - Link to updated manuscript: [https://openreview.net/notes/edits/attachment?id=UshBc3zHQ5&name=pdf](https://openreview.net/notes/edits/attachment?id=UshBc3zHQ5&name=pdf)
>  - Link to manuscript where changes were highlighted: [https://openreview.net/notes/edits/attachment?id=UshBc3zHQ5&name=supplementary_material](https://openreview.net/notes/edits/attachment?id=UshBc3zHQ5&name=supplementary_material)
>
> > **Section 3 needs to be revised. More precisely, the novelty of the proposed approach is not well defined. The section starts by reviewing different transformation methods. Then, thresholding and sparsification/reconstruction methods are employed. Since this pipeline (decomposition, pruning and sparsification) is popularly utilized for data compression, the novelty of the proposed approach should be explained more clearly.**
>
> Thank you for your suggestion. We have added the following text to the Introduction to highlight the novelty of the proposed method:
>
> "Our method is uniquely versatile and capable of handling any combination of layers within the model. This adaptability allows us to optimize the compression process based on the specific characteristics and interactions of the layers. Whether it is a single layer, a complex combination of multiple layers or a whole network, our method consistently delivers efficient and effective compression. This demonstrates the potential of our approach for enhancing the performance and scalability of models across a wide range of applications."
>
> Besides, we highlighted our main contribution by adding:
>
> "Our method stands out in its ability to handle any combination of layers within the model."
>
>
> > **Could you please explain the sparse encoding eqn. (12) and dense decoding eqn. (13) methods?**
>
> Thank you for your question. In order to compute the sparse representation of a tensor, we utilized the [*to_sparse*](https://pytorch.org/docs/stable/generated/torch.Tensor.to_sparse.html) function in PyTorch. It returns a sparse copy of the tensor in the coordinate format (COO).
>
> In contrast, to reconstruct a tensor given a sparse copy of it, we employed the [*to_dense*](https://pytorch.org/docs/stable/generated/torch.Tensor.to_dense.html) function.
>
> To make these things perfectly clear, we will make the code public upon acceptance of the paper.
>
>
> > **In the theoretical analyses (e.g. Theorems), sparsification/reconstruction was not considered. How do they affect the theoretical results?**
>
> Thank you for your comment. In the theoretical analyses, we did consider the sparsification and reconstruction. Specifically, you can find it in Assumption 2 and the Proof of Theorem 1. Please take a look at, for example, Equation 21. We used $\boldsymbol{M}^l$ to make the elements of $\boldsymbol{W}_l \boldsymbol{X}$ sparse. We then employed $\boldsymbol{W}_l^{-1}$ to reconstruct $\boldsymbol{X}^l$. Therefore, the sparsification and reconstruction do not affect the theoretical results. We hope that our explanation will be clear to you.
>
>
> > **How do you employ W of WT in (8) and B of DCT in (9)? Do you further optimize them or use fixed random matrices?**
>
> Thank you for your question. In our work, we fix W of WT in (8) and B of DCT in (9) to the orthogonal matrices as defined by the transforms. We do not further optimize them, as that could affect the reconstruction step.
>
>
> > **Sparse computations on GPU can be computationally complex indeed since they are not supported by many libraries and their implementation can be challenging. Could you please provide implementation details?**
>
> Thank you for your concern. As we mentioned above: to compute the sparse representation and reconstruct a tensor, we utilized [*to_sparse*](https://pytorch.org/docs/stable/generated/torch.Tensor.to_sparse.html) and [*to_dense*](https://pytorch.org/docs/stable/generated/torch.Tensor.to_dense.html) in PyTorch, which uses the native PyTorch implementation of sparse tensors in coordinate (COO) format. To make these things clear, we will make the code public upon acceptance of the paper.

---

> ### Author Response · Authors · 2023-12-06
> **Reply to Reviewer X2os [2]**
>
> > **Could you please provide a detailed footprint analysis of each step in the pipeline?**
>
> Thank you for your suggestion. We have added Figure 4 and a new Section 6.1 where we have discussed the impact on memory footprint.
>
> We have added the following text to the new Section 6.1:
>
> "Figure 4 presents the detailed footprint in each step in the proposed method (see Figure 2) using different settings in small and large arrays with input shape of $128 \times 16 \times 16 \times 1$ and $16 \times 128 \times 128 \times 128 \times 1$, respectively.
>
> Looking at Figure 4, it is apparent that the sparse representation in COO format (see $\boldsymbol{\Pi}$) incurs overhead when the compression ratio is small, *i.e.* $\lambda=1.1$, in both evaluated arrays. This is because very little information is removed in such cases, and the tensors are not remarkably sparse. On the contrary, significant drops in allocated \gls{gpu} memory can be seen from the same figure when $\lambda=10$. A more detailed analysis in terms of $\lambda$ can be found in Section 6.2 and Section 6.6.
>
> The differences between WT and DCT in terms of memory footprint are highlighted in Figure 4. Specifically,
> the allocated memory on GPU of $\boldsymbol{X}$, $\boldsymbol{X}^l$, $\boldsymbol{Y}$ and $\boldsymbol{Y}^l$ of the DCT method are the same except for $\boldsymbol{\Pi}$. By way of contrast, $\boldsymbol{X}$, $\boldsymbol{Y}$, and $\boldsymbol{\Pi}$ of the WT method reserve different amounts of memory. These results are related to the shapes of transformed matrices, $\boldsymbol{Y}$, after the WT and DCT are applied."
>
>
> > **How does reconstruction increase the footprint?**
>
> Thank you for your question. In Figure 9, the reason that reconstruction increases the footprint is due to how PyTorch deal with the ephemeral tensor (NVIDIA CUDA
> Deep Neural Network library (cuDNN) workspace and temporary tensors) and resident buffer (CUDA context, internal tensor fragmentation, and allocator reservation).

---

> ### Author Response · Authors · 2023-12-06
> **Reply to Reviewer X2os [3]**
>
> > **In the motivation, larger DNNs such as transformers have been considered. Could you please extend the analyses with larger ResNets (e.g with layers larger than 50), WideResnets and Transformers?**
>
> Thank you so much for your suggestion. We have now evaluated the proposed method in a wider range of network architectures including ResNet152 with different blocks, WideResNet101, and Vision Transformers in both 2D and 3D. You can see these in the updated Table 2. In addition, we have compared our method to two related works [1,2] and a 16-bit quantization approach (see Table 3).
>
> We have added the following text to Section 6.5.
>
> "Table 3 compares the memory allocated on the GPU using different methods with different parameters.  Instead of employing 32-bit in feature maps and network parameters as in other methods, we analyze the quantization technique with 16-bit. In the “Checkpoint” method (Chen et al., 2016),  s  stands for the number of segments. In WCC, 4-bit weights and 8-bit activations are selected, and $\omega$ denotes the compression rate. In this table, we experiment with our method with different compression ratios,  $\lambda$.
>
> From Table 3, we see the following. First, the 16-bit quantization technique reduces the allocated memory by half of that as in the default configuration *i.e.*  32-bit. For example,  MBL  is 2023 and 1017 in the Efficientnet   B0 when 32-bit and 16-bit are used, respectively. Second, the “Checkpoint” method decreases the GPU usage by large margins in all models except for the combination of DeepLabV3Plus and MobileNetV2 where the sole choice for the number of segments,  $s$, is 1. A possible explanation for this might partly be explained by the fact that the official PyTorch implementation for this method considers MobileNetV2, which is based on inverted residual blocks, as a single module. Third, the WCC consumes more GPU memory than the baseline in all cases. These results are likely to be related to the WCC failed to release ephemeral tensors in the forward pass. Last, in our method, the figures are positive when  $\lambda$  are smaller than 2.5, but become negative afterward. A possible explanation was mentioned earlier that very little is saved when the compression ratio is small, because of the overhead in the sparse representation.
>
> The "Checkpoint" method optimizes GPU memory usage and works well on most evaluated architectures. However, it has a major weakness with architectures based on skip connections (*e.g.*, U-Net) or residual blocks (*e.g.*, ResNet family). To be precise, the "Checkpoint" method fails easily if the feature maps that are checkpointed lie in the middle of residual blocks. In contrast, the proposed method tackles this challenge comfortably and naturally."
>
>  1. Tianqi Chen, Bing Xu, Chiyuan Zhang, and Carlos Guestrin. Training deep nets with sublinear memory cost. Preprint, arXiv:1604.06174, 2016.
>  2. Shahaf E Finder, Yair Zohav, Maor Ashkenazi, and Eran Treister. Wavelet Feature Maps Compression for Image-to-Image CNNs.  Advances in Neural Information Processing Systems, 35, 2022.
>
>
>
> > **Ablations implementing different pruning and sparsification methods should be provided as well.**
>
> Thank you for your comment. Please see the reply to your previous comment. To summarize, we have now included ResNet152, WideResNet101, and Vision Transformers in both 2D and 3D. We also compared the proposed method to existing methods and a 16-bit quantization approach (you can find these in Table 3) on another set of network architectures: Efficientnet, DeepLabV3Plus with MobileNetV2 as the backbone, and EDSR.

---

> ### Author Response · Authors · 2023-12-06
> **Reply to Reviewer X2os [4]**
>
> > **The novelty of the proposed pipeline is not clear. Several implementation and algorithmic details are also missing.**
>
> Thank you for your comment. We have replied to all of your suggested changes above. Here, we summarize what we have updated.
>
>  - The novelty of the proposed method was highlighted in the Introduction: "Our method is uniquely versatile and capable of handling any combination of layers within the model. This adaptability allows us to optimize the compression process based on the specific characteristics and interactions of the layers. Whether it is a single layer, a complex combination of multiple layers or a whole network, our method consistently delivers efficient and effective compression. This demonstrates the potential of our approach for enhancing the performance and scalability of models across a wide range of applications."
>  - Main contribution was also added the follows: "Our method stands out in its ability to handle any combination of layers within the model."
>  - Implementation of sparse and dense representations were detailed.
>  - Sparsification and reconstruction were considered in the Theorems. Thus, they do not affect the theoretical results.
>  - In our work, we fix W of WT in (8) and B of DCT in (9) to the orthogonal matrices as defined by the transforms. We do not further optimize them, as that could affect the reconstruction step.
>
> Besides, we have added the following text to the Section 3.1.2:
>
> Let $\boldsymbol{X} \in \mathbb{R}^{2m \times 2m}$ be a real array, which is a split block. In this case, $m$ is equivalent to $8$. The DCT of $\boldsymbol{X}$ is defined as,
>
> $$y_{i,j}
>         =
>             \frac{1}{m}
>             \sum_{u=0}^{2m-1}
>             \sum_{v=0}^{2m-1}
>             \Lambda_i
>             \Lambda_j
>             \cdot
>             \mathrm{cos} \frac{(2u+1) \pi i}{4m}
>         %     \nonumber
>         %     \\
>         % & \quad
>             \cdot
>             \mathrm{cos} \frac{(2v+1) \pi j}{4m}
>             \cdot
>             x_{u,v},
> $$
>
> with $x_{u,v} \in \boldsymbol{X}$ and $y_{i,j} \in \boldsymbol{Y}$. The $u$ and $i$ denote the row index of $\boldsymbol{X}$ and $\boldsymbol{Y}$, respectively. The $v$ and $j$ denote the column index of $\boldsymbol{X}$ and $\boldsymbol{Y}$, respectively. The $\Lambda_k$ is
> $$
>     \Lambda_k = \begin{cases}
>                     \frac{1}{\sqrt{2}}, & \text{if~} k=0, \\
>                     1,   & \text{otherwise}.
>                 \end{cases}
> $$
>
> The corresponding inverse \gls{2D} \gls{dct} is defined,
> $$
>     x_{u,v}
>         =
>             \frac{1}{m}
>             \sum_{i=0}^{2m-1}
>             \sum_{j=0}^{2m-1}
>             \Lambda_i
>             \Lambda_j
>             \cdot
>             \mathrm{cos} \frac{(2u+1) \pi i}{4m}
>             \cdot
>             \mathrm{cos} \frac{(2v+1) \pi j}{4m}
>             \cdot
>             y_{i,j},
> $$
> with $x_{u,v} \in \boldsymbol{X}$ and $y_{i,j} \in \boldsymbol{Y}$.
>
>
>
> > **The theoretical analyses should be improved considering all of the proposed methods, e.g. including sparsification.**
>
> Thank you for your comment. In the theoretical analyses, we did consider the sparsification and reconstruction. Specifically, you can find it in Assumption 2 and the Proof of Theorem 1. Please take a look at, for example, Equation 21. We used $\boldsymbol{M}^l$ to make the elements of $\boldsymbol{W}_l \boldsymbol{X}$ sparse. We then employed $\boldsymbol{W}_l^{-1}$ to reconstruct $\boldsymbol{X}^l$. Therefore, the sparsification and reconstruction do not affect the theoretical results. We hope that our explanation will be clear to you.
>
>
>
> > **Experimental analyses should be improved using additional models and with ablations.**
>
> Thank you for your comment. We have performed experimental analyses as suggested.
>
> To summarize, we have now included ResNet152, WideResNet101, and Vision Transformers in both 2D and 3D. We also compared the proposed method to existing methods and a 16-bit quantization approach (you can find these in Table 3) on another set of network architectures: Efficientnet, DeepLabV3Plus with MobileNetV2 as the backbone, and EDSR.

---

### Review · Reviewer_MXoD · 2023-11-09

**Summary Of Contributions:**

The paper addresses the challenge of dealing with large-scale model parameters that cannot fit into standard GPUs.
• The authors propose an innovative approach to compress high-dimensional activation maps, which are the most memory-intensive components in modern deep learning model training.
• The proposed method involves saving GPU memory by storing compressed activation maps during forward propagation and using their corresponding decompressed versions during backward propagation for gradient computation.
• Three compression techniques, namely Wavelet Transform, Discrete Cosine Transform, and Simple Thresholding, are employed.
• The authors conducted experiments on two classification tasks and two semantic segmentation tasks, primarily using ResNet-18 and U-NET architectures across various compression ratios and model layers, achieving up to a 95% reduction in memory usage.
• The authors also claim that their approach induces a regularization effect on layer weight gradients.

**Audience:**

Yes

**Claims And Evidence:**

Yes

**Requested Changes:**

The paper should explain why the compressed and thresholded variables are not used in back-propagation and what challenges this may pose.
• Instead of invoking Lemma 1, it might be clearer to specify that the same approach used in Lemma 1 should be employed for computing expectations in the proof of Corollary 1.
• Thepapershouldinvestigateandexplainwhytheaccuracycurvedecreases slowly from block 3 to 4 for λ = 2.5 and why Figure 4b shows a significant drop in model accuracy.
• The authors need to examine and elucidate the unexpected behavior mentioned in Table 2, where compressing more layers results in more GPU memory savings for U-NET but not for ResNet-18.

**Strengths And Weaknesses:**

Strengths:

The paper effectively underscores the importance of the problem, providing quantitative data on the memory requirements of different deep neural network models.
• It highlights the necessity for compressing activation maps in deep neural networks to address their overparameterization.
• The paper thoroughly discusses memory requirements at different stages of the training process.
• The work is grounded in solid theoretical foundations, particularly Lemma 1 and Theorem 1.
• The experimental results are generally satisfactory, and the authors conduct a comprehensive ablation study to assess the impact of their compression method on various aspects, including training accuracy, image reconstruction, GPU memory allocation, training speed, and to some extent, regularization.

Weaknesses:

The paper should clarify the distinction and purpose of Corollary 5, 6, and 7 compared to Corollary 2, 3, and 4.
• It should provide a more comprehensive discussion of how their method differs from and outperforms existing lossless compression techniques mentioned in related work.
• The handling of the parameter λ needs more explanation, especially in cases with a significant number of high or low values in the array. The paper could provide examples to illustrate this issue.
• The use of thresholding techniques and the relevance of Assumption 1 should be more convincingly justified.
• The paper lacks insights into the applicability of the proposed method to complex architectures beyond ResNet-18 and U-NET. It should explore the potential impact on model performance and GPU memory allocation for various modern architectures.

---

> ### Author Response · Authors · 2023-12-06
> **Reply to Reviewer MXoD [1]**
>
> - Link to updated manuscript: [https://openreview.net/notes/edits/attachment?id=UshBc3zHQ5&name=pdf](https://openreview.net/notes/edits/attachment?id=UshBc3zHQ5&name=pdf)
>  - Link to manuscript where changes were highlighted: [https://openreview.net/notes/edits/attachment?id=UshBc3zHQ5&name=supplementary_material](https://openreview.net/notes/edits/attachment?id=UshBc3zHQ5&name=supplementary_material)
>
> > **The paper should clarify the distinction and purpose of Corollary 5, 6, and 7 compared to Corollary 2, 3, and 4.**
>
> Thank you so much for spotting the mistake. They are actually the same, and we left them unintentionally when moving things around in the manuscript. We have now removed Corollaries 5, 6, and 7 in the updated paper. Again, thank you!
>
>
> > **It should provide a more comprehensive discussion of how their method differs from and outperforms existing lossless compression techniques mentioned in related work.**
>
> Thank you for this comment. We have now compared our method to several methods that you can find in the new Table 3. We have added the following text to Section 6.6 as well.
>
> "Table 3 compares the memory allocated on the GPU using different methods with different parameters.  Instead of employing 32-bit in feature maps and network parameters as in other methods, we analyze the quantization technique with 16-bit. In the “Checkpoint” method (Chen et al., 2016),  s  stands for the number of segments. In WCC, 4-bit weights and 8-bit activations are selected, and $\omega$ denotes the compression rate. In this table, we experiment with our method with different compression ratios,  $\lambda$.
>
> From Table 3, we see the following. First, the 16-bit quantization technique reduces the allocated memory by half of that as in the default configuration *i.e.*  32-bit. For example,  MBL  is 2023 and 1017 in the Efficientnet   B0 when 32-bit and 16-bit are used, respectively. Second, the “Checkpoint” method decreases the GPU usage by large margins in all models except for the combination of DeepLabV3Plus and MobileNetV2 where the sole choice for the number of segments,  $s$, is 1. A possible explanation for this might partly be explained by the fact that the official PyTorch implementation for this method considers MobileNetV2, which is based on inverted residual blocks, as a single module. Third, the WCC consumes more GPU memory than the baseline in all cases. These results are likely to be related to the WCC failed to release ephemeral tensors in the forward pass. Last, in our method, the figures are positive when  $\lambda$  are smaller than 2.5, but become negative afterward. A possible explanation was mentioned earlier that very little is saved when the compression ratio is small, because of the overhead in the sparse representation.
>
> The "Checkpoint" method optimizes GPU memory usage and works well on most evaluated architectures. However, it has a major weakness with architectures based on skip connections (*e.g.*, U-Net) or residual blocks (*e.g.*, ResNet family). To be precise, the "Checkpoint" method fails easily if the feature maps that are checkpointed lie in the middle of residual blocks. In contrast, the proposed method tackles this challenge comfortably and naturally."
>
>
> > **The handling of the parameter $\lambda$ needs more explanation, especially in cases with a significant number of high or low values in the array. The paper could provide examples to illustrate this issue.**
>
> Thank you for your suggestion. We have added Figure 3 to illustrate how $\lambda$ handles three different cases: the first array has a substantial number of low values, the second has a normal-like distribution, and the third has a substantial number of high values.

---

> > ### Author Response · Authors · 2023-12-06
> > **Reply to Reviewer MXoD [2]**
> >
> > > **The use of thresholding techniques and the relevance of Assumption 1 should be more convincingly justified.**
> >
> > Thank you for your comment. Our thresholding techniques are very common both in the literature and in practical standards. They are based on the discrete cosine transform (DCT) and the discrete wavelet transform (DWT), which are used in common compressed image formats, such as JPEG and JPEG 2000, and in compressed video formats, such as MPEG-C, respectively.
> >
> > Assumption 1 is introduced in order to address that elements disappear through the mask array, $\boldsymbol{M}^l$. We model this pixel dropout using a Bernoulli distribution with probability $1/\lambda$. This assumption is then used to prove Lemma 1.
> >
> >
> > > **The paper lacks insights into the applicability of the proposed method to complex architectures beyond ResNet-18 and U-NET. It should explore the potential impact on model performance and GPU memory allocation for various modern architectures.**
> >
> > Thank you for your comment. Please see my reply to your previous comment. To summarize, we have now included ResNet152, WideResNet101, and Vision Transformers in both 2D and 3D. We also compared our proposed method to existing methods and a 16-bit quantization approach (you can find it in Table 3) on another set of network architectures: Efficientnet, DeepLabV3Plus with MobileNetV2 as the backbone, and EDSR.
> >
> >
> > > **The paper should explain why the compressed and thresholded variables are not used in back-propagation and what challenges this may pose.**
> >
> > Thank you for this comment. We actually do use the compressed tensor in the back-propagation, when computing the gradient, and it is used instead of the original (not compressed) tensor. You can see the illustration of the proposed compression method in Figure 2.
> >
> > If our explanations or the figure are confusing, please let us know. We would be happy to clarify this further.
> >
> >
> > > **Instead of invoking Lemma 1, it might be clearer to specify that the same approach used in Lemma 1 should be employed for computing expectations in the proof of Corollary 1.**
> >
> > Thank you for your suggestion. We have updated our manuscript accordingly.
> >
> >
> > > **The paper should investigate and explain why the accuracy curve decreases slowly from block 3 to 4 for $\lambda = 2.5$ and why Figure 4b shows a significant drop in model accuracy.**
> >
> > Thank you for your comment. Please note that we have added two more figures, so that the previous Figure 4 now becomes Figure 6.
> >
> > We have added the following discussion to the manuscript:
> >
> > "There are several possible explanations for these results. First, the resolutions of features maps in the third block and last block of ResNet18 are $256 \times 8 \times 8$ and $512 \times 4 \times 4$, respectively. Therefore, when $\lambda$ is larger than 2.5, very few details are left causing the network to fail to learn (see "Block3" and "Block4" in Figure 6b). Second, when all blocks are compressed, the information, that is carried from prior blocks ("Block0" to "Block2") might be insufficient to make the last two compressed blocks ("Block3" and "Block4") extract relevant features. In other words, this may be explained by collective information loss."
> >
> >
> > > **The authors need to examine and elucidate the unexpected behavior mentioned in Table 2, where compressing more layers results in more GPU memory savings for U-NET but not for ResNet-18.**
> >
> > Thank you very much for your comment. We have added the following explanation to the manuscript:
> >
> > "The reason for this unexpected behavior is likely due to the presence of addition operation in ResNet block that prevents the original tensor to be released from GPU memory *i.e.* PyTorch stores both original and compressed tensors on the GPU."

---

> > ### Comment · Reviewer_MXoD · 2024-01-04
> > **Response to the revison.**
> >
> > I thank the Authors for addressing my comments. I am satisfied with their response.

---

### Review · Reviewer_gCkT · 2023-11-22

**Summary Of Contributions:**

The paper presents anadvancement in addressing the memory consumption issues in deep learning, particularly for GPU-based architectures. Its strengths lie in its innovative approach and comprehensive testing, but there are areas where further research and elaboration could strengthen its contributions, particularly regarding practical implementation and performance implications.

**Audience:**

Yes

**Claims And Evidence:**

Yes

**Requested Changes:**

Look at weaknesses

**Strengths And Weaknesses:**

Strengths
+The proposed method for compressing high-dimensional activation maps addresses a significant challenge in deep learning - the large memory consumption of contemporary deep learning architectures, particularly in GPUs​​.

+The paper not only introduces a method for compression but also demonstrates that this method induces a regularization effect on the layer weight gradients. This dual effect (compression and regularization) adds significant value to the approach​​.

+The regularizing effect of the proposed method helps in reducing overfitting, which is a common challenge in deep learning models, especially those with large numbers of parameters​​.

+ The paper gives a good overview of related work in the field, providing context for the significance and novelty of its contribution​​.

Weaknesses
-While the paper outlines the method's theoretical basis, the practical challenges of implementing this approach in diverse real-world scenarios might be underrepresented. The complexity of the method, especially in terms of the mathematical formulations and assumptions involved, could pose implementation challenges​​. Can the authors comment on how their infrastructure is pytorch or tensorflow compatible? Does it add any overhead to the training?

- While the paper discusses the regularization effect, it could explore more deeply how the compression affects the overall performance of the neural network, especially in terms of accuracy and loss, beyond just the reduction in memory usage.

- The evaluation, though comprehensive, seems to focus more on memory reduction and regularizing effects. An expanded evaluation including a wider range of neural network architectures and applications could provide a more rounded assessment of the method's versatility and limitations.

- The effectiveness of the method in various hardware configurations is not deeply explored. Given that the activation map compression is particularly relevant for GPU memory optimization, its performance across different GPU architectures and configurations would be valuable.

-A more detailed comparative analysis with existing methods, particularly in terms of trade-offs between compression, performance, and training efficiency, would provide a clearer understanding of the method's relative advantages and disadvantages.

---

> ### Author Response · Authors · 2023-12-06
> **Reply to Reviewer gCkT [1]**
>
> - Link to updated manuscript: [https://openreview.net/notes/edits/attachment?id=UshBc3zHQ5&name=pdf](https://openreview.net/notes/edits/attachment?id=UshBc3zHQ5&name=pdf)
>  - Link to manuscript where changes were highlighted: [https://openreview.net/notes/edits/attachment?id=UshBc3zHQ5&name=supplementary_material](https://openreview.net/notes/edits/attachment?id=UshBc3zHQ5&name=supplementary_material)
>
> > **While the paper outlines the method's theoretical basis, the practical challenges of implementing this approach in diverse real-world scenarios might be underrepresented. The complexity of the method, especially in terms of the mathematical formulations and assumptions involved, could pose implementation challenges. Can the authors comment on how their infrastructure is pytorch or tensorflow compatible? Does it add any overhead to the training?**
>
> Thank you very much for your feedback. To make the proposed method perfectly clear, we will make the code public upon acceptance of the paper.
>
> As you can see in the section on "Implementation Details and Training", the proposed method was implemented in PyTorch. We believe that an implementation in TensorFlow would be straightforward, and it adds very little overhead to the training since the WT has linear asymptotic complexity and the DCT has log-linear asymptotic complexity (the complexity of WT is like that of a convolution and the complexity of DCT is like that of the fast Fourier transform).
>
>
> > **While the paper discusses the regularization effect, it could explore more deeply how the compression affects the overall performance of the neural network, especially in terms of accuracy and loss, beyond just the reduction in memory usage.**
>
> We have investigated the impact of the proposed activation compression not only on memory usage reduction but also on the following factors:
>
>  - Memory Footprint Impact in Section 6.1.
>  - Reconstructed Image Impact in Section 6.2.
>  - Performance Impact in Section 6.3 where you can find the effects in terms of accuracy for the classification task and dice coefficient for the segmentation task.
>  - Regularization Impact in Section 6.4.
>  - Training Speed Impact in Section 6.5.

---

> ### Author Response · Authors · 2023-12-06
> **Reply to Reviewer gCkT [2]**
>
> > **The evaluation, though comprehensive, seems to focus more on memory reduction and regularizing effects. An expanded evaluation including a wider range of neural network architectures and applications could provide a more rounded assessment of the method's versatility and limitations.**
>
> Thank you so much for your suggestion. We have now evaluated our proposed method in a wider range of network architectures including ResNet152 with different blocks, WideResNet101, and Vision Transformers in both 2D and 3D. You can see these in the updated Table 2. In addition, we have compared our method to two related works [1,2] and a 16-bit quantization approach (see Table 3).
>
> We have added the following text to Section 6.5:
>
> "Table 3 compares the memory allocated on the GPU using different methods with different parameters.  Instead of employing 32-bit in feature maps and network parameters as in other methods, we analyze the quantization technique with 16-bit. In the “Checkpoint” method (Chen et al., 2016),  s  stands for the number of segments. In WCC, 4-bit weights and 8-bit activations are selected, and $\omega$ denotes the compression rate. In this table, we experiment with our method with different compression ratios,  $\lambda$.
>
> From Table 3, we see the following. First, the 16-bit quantization technique reduces the allocated memory by half of that as in the default configuration *i.e.*  32-bit. For example,  MBL  is 2023 and 1017 in the Efficientnet   B0 when 32-bit and 16-bit are used, respectively. Second, the “Checkpoint” method decreases the GPU usage by large margins in all models except for the combination of DeepLabV3Plus and MobileNetV2 where the sole choice for the number of segments,  $s$, is 1. A possible explanation for this might partly be explained by the fact that the official PyTorch implementation for this method considers MobileNetV2, which is based on inverted residual blocks, as a single module. Third, the WCC consumes more GPU memory than the baseline in all cases. These results are likely to be related to the WCC failed to release ephemeral tensors in the forward pass. Last, in our method, the figures are positive when  $\lambda$  are smaller than 2.5, but become negative afterward. A possible explanation was mentioned earlier that very little is saved when the compression ratio is small, because of the overhead in the sparse representation.
>
> The "Checkpoint" method optimizes GPU memory usage and works well on most evaluated architectures. However, it has a major weakness with architectures based on skip connections (*e.g.*, U-Net) or residual blocks (*e.g.*, ResNet family). To be precise, the "Checkpoint" method fails easily if the feature maps that are checkpointed lie in the middle of residual blocks. In contrast, the proposed method tackles this challenge comfortably and naturally."
>
>  1. Tianqi Chen, Bing Xu, Chiyuan Zhang, and Carlos Guestrin. Training deep nets with sublinear memory cost. Preprint, arXiv:1604.06174, 2016.
>  2. Shahaf E Finder, Yair Zohav, Maor Ashkenazi, and Eran Treister. Wavelet Feature Maps Compression for Image-to-Image CNNs.  Advances in Neural Information Processing Systems, 35, 2022.
>
>
> > **The effectiveness of the method in various hardware configurations is not deeply explored. Given that the activation map compression is particularly relevant for GPU memory optimization, its performance across different GPU architectures and configurations would be valuable.**
>
> Thank you for this comment. In addition to ResNet18 and U-Net, we have now included ResNet152, WideResNet101, and Vision Transformer 2D/3D. We also compared our method to the existing methods and a 16-bit quantization approach (you can find it in Table 3) on another set of network architectures including Efficientnet, DeepLabV3Plus with MobileNetV2 as the backbone, and EDSR.
>
>
> > **A more detailed comparative analysis with existing methods, particularly in terms of trade-offs between compression, performance, and training efficiency, would provide a clearer understanding of the method's relative advantages and disadvantages.**
>
> Thank you for your comment. We, indeed, have investigated the impact of the proposed activation compression not only on memory usage reduction but also on the following factors:
>
>  - Memory Footprint Impact in Section 6.1.
>  - Reconstructed Image Impact in Section 6.2.
>  - Performance Impact in Section 6.3 where you can find the effects in terms of accuracy for the classification task and dice coefficient for the segmentation task.
>  - Regularization Impact in Section 6.4.
>  - Training Speed Impact in Section 6.5.
>
> In addition to ResNet18 and U-Net, we have now also included numerous other network architectures: ResNet152, WideResNet101, Vision Transformer 2D/3D, Efficientnet, DeepLabV3Plus with MobileNetV2 as the backbone, and EDSR.

---

> > ### Comment · Reviewer_gCkT · 2024-01-05
> > **Reviewer Response**
> >
> > All my comments have been adequately addressed.

---

### Author Response · Authors · 2024-03-10
**Camera-Ready Version**

Dear Action Editor and Reviewers,

We are pleased to submit the camera-ready version of our manuscript, "Compressing the Activation Maps in Deep Neural Networks and Its Regularizing Effect". We would like to express our sincere gratitude to the reviewers and the action editor for their valuable comments and suggestions, which significantly improved the quality of our work.

Sincerely,
Authors

---

### Decision · Action_Editor_bPBz · 2024-01-25

**Recommendation:** Accept as is

**Comment:**

The AE sides with acceptance. This work provides an experimental study of wavelet, DCT, and thresholding transforms for compressing features during training across a variety of deep network architectures for vision on MNIST, CIFAR-100, BRATS19, and SPLEEN. While these are known transforms, the results here are carefully measured and informative, and several categories of related work is credited (though please see the concluding note for a suggestion). The review and revision process has resulted in clearer text and expanded results to cover a greater diversity of model architectures.

The AE thanks the authors and reviewers for engaging in the TMLR process to deliver an improved paper with agreement between its claims and evidence and a clear audience.

Feedback to authors:

- The AE suggests citing a survey on model compression in the related work due to the popularity of the topic and the notes by reviewers on the existence of prior methods. For instance, please consider the survey [Model Compression and Hardware Acceleration for Neural Networks: A Comprehensive Survey](https://ieeexplore.ieee.org/abstract/document/9043731) by Deng et al. in IEEE'20. A pointer to this is a positive, while a highlighting of related methods on transforming activations is better still.
- The AE suggests reformatting to avoid the need to rotate Table 3 on Page 26 for readability, by for example by distributing the architecture columns across multiple rows.

**Audience:**

All reviewers agree that there is an audience. The audience for this work includes those studying and applying deep learning computation on memory-constrained platforms, the intersection of image signal processing and deep learning, and regularization for the optimization of deep networks for vision (convolutional networks and ViTs). This work applies to activations/intermediate representations from a single layer to an entire network for relevance across different model and experiment sizes. The experimental scope is restricted to toy image classification datasets (MNIST, CIFAR-100) and small-scale medical segmentation datasets (BRATS19 and SPLEEN with 335 and 41 images, respectively), which narrows the audience, but within this scope the measurements are comprehensive and so can inform an audience interested in this scale.

**Claims And Evidence:**

All reviewers agree that the claims and evidence are connected and satisfactory. Reviewers confirm the adequacy of the experiments as an applied study, and support the description of the experiments in the abstract and throughout the text, but note limitations w.r.t. scale, the existence of prior methods, and the library-specific nature of the computational findings for PyTorch. Each reviewer is satisfied by the revisions made following submission. In the claimed scope of datasets and models, the experimental results encompass a variety of measures (accuracy, memory, regularization, reconstruction quality, and training speed), so although the scope is narrow it is thoroughly covered. As a last dimension of evidence, the authors agree to release the code on acceptance to further detail the experiments and confirm the results.